# Homeodomain proteins hierarchically specify neuronal diversity and synaptic connectivity

Chundi Xu[1]*, Tyler B Ramos[1], Edward M Rogers[2], Michael B Reiser[2], Chris Q Doe[1]

[1]Institute of Neuroscience, Howard Hughes Medical Institute, University of Oregon, Eugene, United States; [2]Janelia Research Campus, Howard Hughes Medical Institute, Helix Drive, Ashburn, United States

*For correspondence:
cxu3@uoregon.edu

Competing interest: The authors declare that no competing interests exist.

**Abstract** How our brain generates diverse neuron types that assemble into precise neural circuits remains unclear. Using *Drosophila* lamina neuron types (L1-L5), we show that the primary homeodomain transcription factor (HDTF) brain-specific homeobox (Bsh) is initiated in progenitors and maintained in L4/L5 neurons to adulthood. Bsh activates secondary HDTFs Ap (L4) and Pdm3 (L5) and specifies L4/L5 neuronal fates while repressing the HDTF Zfh1 to prevent ectopic L1/L3 fates (control: L1-L5; Bsh-knockdown: L1-L3), thereby generating lamina neuronal diversity for normal visual sensitivity. Subsequently, in L4 neurons, Bsh and Ap function in a feed-forward loop to activate the synapse recognition molecule DIP-β, thereby bridging neuronal fate decision to synaptic connectivity. Expression of a Bsh:Dam, specifically in L4, reveals Bsh binding to the *DIP-β* locus and additional candidate L4 functional identity genes. We propose that HDTFs function hierarchically to coordinate neuronal molecular identity, circuit formation, and function. Hierarchical HDTFs may represent a conserved mechanism for linking neuronal diversity to circuit assembly and function.

## eLife assessment

This paper, offering insights into the mechanisms of neuronal cell type diversification, provides **important** findings that have theoretical or practical implications beyond a single subfield. The data are **compelling** and provide evidence that features methods, data and analyses that are more rigorous than the current state-of-the-art.

## Introduction

Our ability to perceive and respond to the world requires a diverse array of neuron types characterized initially by transcription factor (TF) combinatorial codes, followed by neuron-type-specific functional attributes such as cell surface molecules, neurotransmitters, and ion channels. It has been well documented how initial neuronal diversity is generated: in both *Drosophila* and mouse, spatial and temporal factors act combinatorially to generate molecularly distinct newborn neurons (*Bayraktar and Doe, 2013*; *Doe, 2017*; *Erclik et al., 2017*; *Holguera and Desplan, 2018*; *Sen et al., 2019*). Yet, most spatial and temporal factors are only transiently present in newborn neurons; therefore, another mechanism is required to bridge initial fate to mature features such as connectivity, neurotransmitters, and ion channels. It remains poorly understood how the initial fate decision of newborn neurons leads to the functional identity of mature neurons. Work from the Hobert lab in *Caenorhabditis elegans* found that each adult neuron type expresses a unique combination of homeodomain TFs (HDTFs), which have been called terminal selectors (*Hobert, 2021*; *Reilly et al., 2020*). Terminal selector HDTFs not only drive the expression of neuron functional identity genes but also activate pan-neuronal genes

(*Hobert, 2021*; *Howell et al., 2015*; *Kratsios et al., 2015*; *Stefanakis et al., 2015*). Loss of terminal selector HDTFs frequently results in altered neuronal identity and function (*Arlotta and Hobert, 2015*; *Cros and Hobert, 2022*; *Reilly et al., 2022*). Although a great deal is known about HDTFs in the specification of neuron-type-specific morphology and synaptic connectivity in *C. elegans* (*Chen et al., 2012*; *Cubelos et al., 2010*; *De Marco Garcia and Jessell, 2008*; *Friedrich et al., 2016*; *Hasegawa et al., 2013*; *Özel et al., 2022*; *Sakkou et al., 2007*; *Santiago and Bashaw, 2014*; *Thor and Thomas, 1997*), it remains unknown the extent to which this model is generalizable to other organisms. Here we use the *Drosophila* lamina, the first ganglion in the optic lobe, to test the hypothesis that HDTFs couple the initial fate decision to later circuit formation and functional aspects of the neuron.

The *Drosophila* lamina has only five intrinsic neuron types (L1-L5), which are analogous to bipolar cells in the vertebrate (*Sanes and Zipursky, 2010*). During late larval and early pupal stages, lamina progenitor cells (LPCs) give rise to L1-L5 neurons (*Fernandes et al., 2017*; *Huang et al., 1998*). The core motion detection neurons L1-L3 receive direct synaptic inputs from photoreceptors and mediate visual motion detection, whereas L4 and L5 receive synaptic inputs from L2 and L1, respectively, and their function is currently unclear (*Meinertzhagen and O'Neil, 1991*; *Rivera-Alba et al., 2011*; *Silies et al., 2013*; *Tuthill et al., 2013*). The cell bodies of each lamina neuron type are localized in a layer-specific manner (*Tan et al., 2015*). L2/L3 cell bodies are intermingled in the most distal layer, while L1, L4, and L5 form distinct layers progressively more proximal (*Figure 1*). Each lamina neuron type expresses unique TF markers. L1, L2, and L3 neurons express Svp, Bab2, and Erm, respectively (*Tan et al., 2015*). L4 and L5 neurons express the HDTFs Bsh/Ap and Bsh/Pdm3, respectively (*Hasegawa et al., 2013*; *Tan et al., 2015*). Work from Hasegawa et al. has shown that in *bsh* mutants, L4 adopts L3-like morphology, L5 becomes glia, and Ap mRNA level is reduced (*Hasegawa et al., 2013*). This suggests that the HDTF Bsh is important for L4/L5 neuron-type specification.

Here, we show that HDTFs function hierarchically in coupling neuronal fate specification to circuit assembly. First, a primary (earlier-initiated) HDTF Bsh activates two secondary (later-initiated) HDTFs Ap (L4) and Pdm3 (L5) and specifies L4 and L5 neuronal fates. Next, we show that the primary HDTF Bsh generates lamina neuronal diversity (control: L1-L5; Bsh-knockdown [KD]: L1-L3) required for normal visual sensitivity. Third, in L4 neurons, Bsh and Ap function in a feed-forward loop to activate the synapse recognition molecule (DIP-β), thereby bridging neuronal fate decision to synaptic connectivity. Fourth, Bsh in L4 neurons directly binds at the DIP-β locus and other L4 functional identity genes. Our work may provide a conserved mechanism of HDTFs hierarchically linking neuronal diversity to circuit assembly.

## Results

### Sequential expression of HDTFs during lamina neurogenesis

L4 and L5 neurons are generated by a subset of LPCs during late larval and early pupal stages (*Fernandes et al., 2017*; *Huang et al., 1998*), but it is unknown exactly when Bsh, Ap, and Pdm3 are initially expressed. To address this question, we identified the Tailless (Tll) TF as a novel marker for LPCs (*Figure 1A–C*). Indeed, Tll and the neuronal marker Elav have precise complementary high-level expression, validating Tll as an LPC marker (*Figure 1A–C*, summarized in *Figure 1G*). Importantly, we found that Bsh was first detected in a subset of Tll+ LPCs (*Figure 1C*). In contrast, Ap and Pdm3 were first detected much later in L4 and L5 neurons, respectively (*Figure 1D–E*). Ap and Pdm3 were never detected in LPCs or newborn L4 or L5 neurons (*Figure 1D–E*), showing that Bsh is expressed prior to Ap and Pdm3. Expression of Bsh, Ap, and Pdm3 was maintained in L4/L5 neurons into the adult (*Figure 1F*), consistent with a potential role as terminal selectors in maintaining neuron identity (*Deneris and Hobert, 2014*; *Serrano-Saiz et al., 2018*). We conclude that Bsh expression is initiated in LPCs, while Ap and Pdm3 are initiated in neurons. Due to its earlier expression, we refer to Bsh as a primary HDTF, and due to their later expression, we refer to Ap and Pdm3 as secondary HDTFs (*Figure 1H*).

### Bsh activates Ap/Pdm3 expression and specifies L4/L5 neuronal fates

Bsh is expressed prior to Ap and Pdm3, raising the possibility that Bsh activates Ap and Pdm3 expression. We used R27G05-Gal4 to express Bsh-RNAi in LPCs. To confirm whether R27G05-Gal4 is turned on in LPCs, we used R27G05-Gal4>UAS-myristoylated-GFP, and indeed, GFP can be detected in all

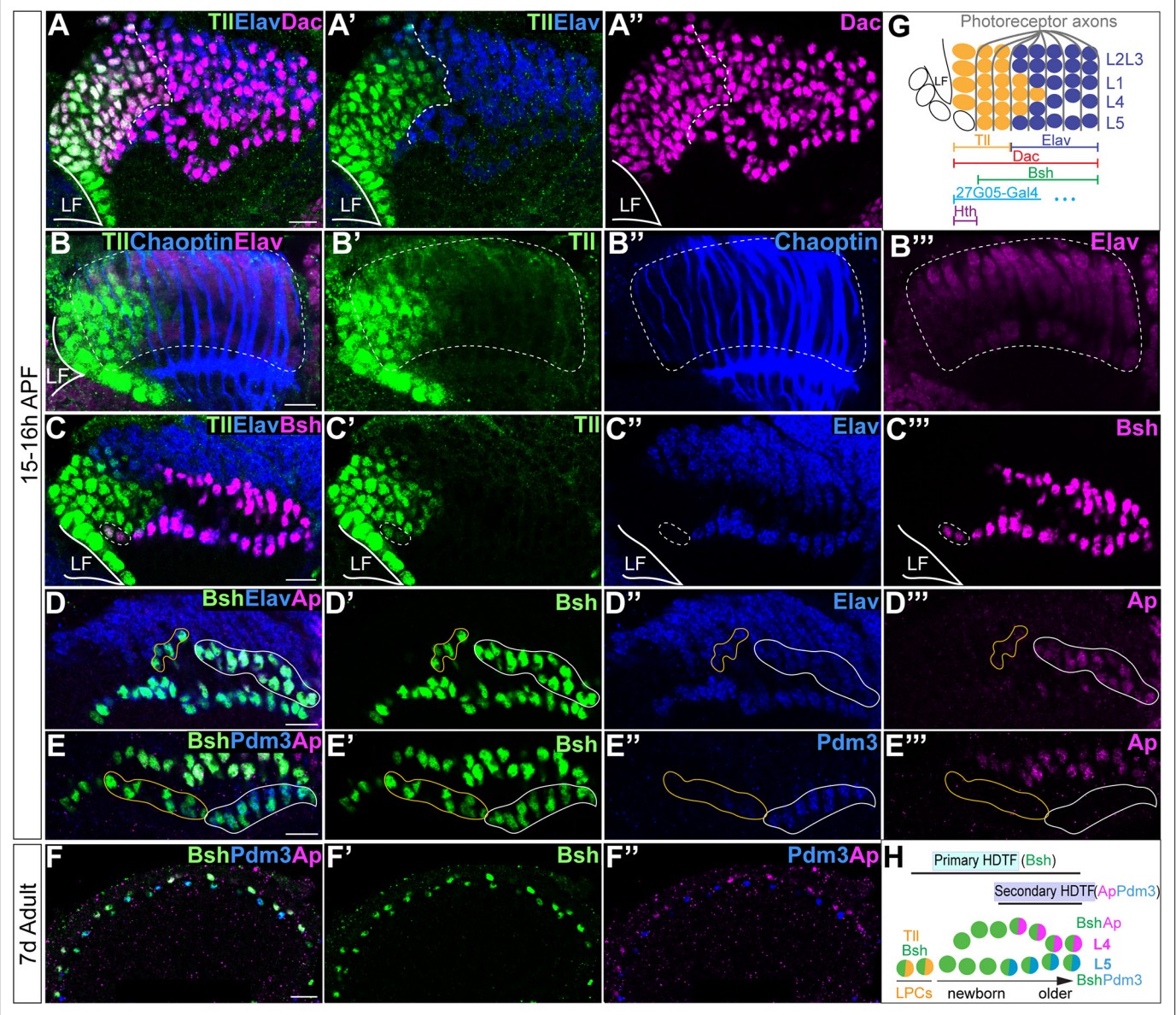

**Figure 1.** Sequential initiation of homeodomain transcription factors (HDTFs) during lamina neurogenesis. (**A–A‴**) Tll is identified as a lamina progenitor cell (LPC) marker, expressed complementary to Elav; Dac labels both Tll+ LPCs and Elav+ neurons. LF: lamina furrow. Here and below, scale bar: 10 μm, n≥5 brains. (**B–B‴**) Tll+ cells are localized both within the lamina columns and before the columns. Lamina columns (white dash circle) are outlined by the photoreceptor axons, which are labeled by Chaoptin. n≥5 brains. LF: lamina furrow. (**C–C‴**) Bsh is expressed in Tll+ Elav LPCs (white dash circle) as well as in Elav+ L4 and L5 neurons. (**D–D‴**) Ap is expressed in L4 neurons. Newborn L4 neurons are Bsh+ Elav+ Ap- (yellow line circle), while older L4 neurons are Bsh+ Elav+ Ap+ (white line circle). (**E–E‴**) Pdm3 is expressed in L5 neurons. Newborn L5 neurons are Bsh+ Pdm3- (yellow line circle), while older L5 neurons are Bsh+ Pdm3+ (white line circle). (**F–F‴**) Bsh, Ap, and Pdm3 expressions are maintained in adults. (**G**) Schematic of lamina neuron development in early pupa. (**H**) Summary.

LPCs and lamina neurons until ~66 hr after pupa formation (APF) (*Figure 2—figure supplement 1A, D, E*). To test whether R27G05-Gal4 can be turned on in lamina neurons, we took advantage of tubP-Gal80[ts] to temporally control R27G05-Gal4>UAS-myristoylated-GFP. We found that GFP can be detected in most lamina neurons when tubP-Gal80[ts] was inactivated from the beginning of lamina neurogenesis. In contrast, GFP is absent in most lamina neurons when tubP-Gal80[ts] was inactivated from the end of lamina neurogenesis, suggesting that R27G05-Gal4 is functionally an LPC-Gal4 trans-gene (*Figure 2—figure supplement 1B, C*). As expected, Bsh-KD (R27G05-Gal4>UAS-Bsh-RNAi) eliminates Bsh expression in LPCs and newborn neurons (*Figure 2A–F*). Bsh remains undetectable in the lamina neurons in the adult, despite the lack of RNAi expression, which indicates that Bsh

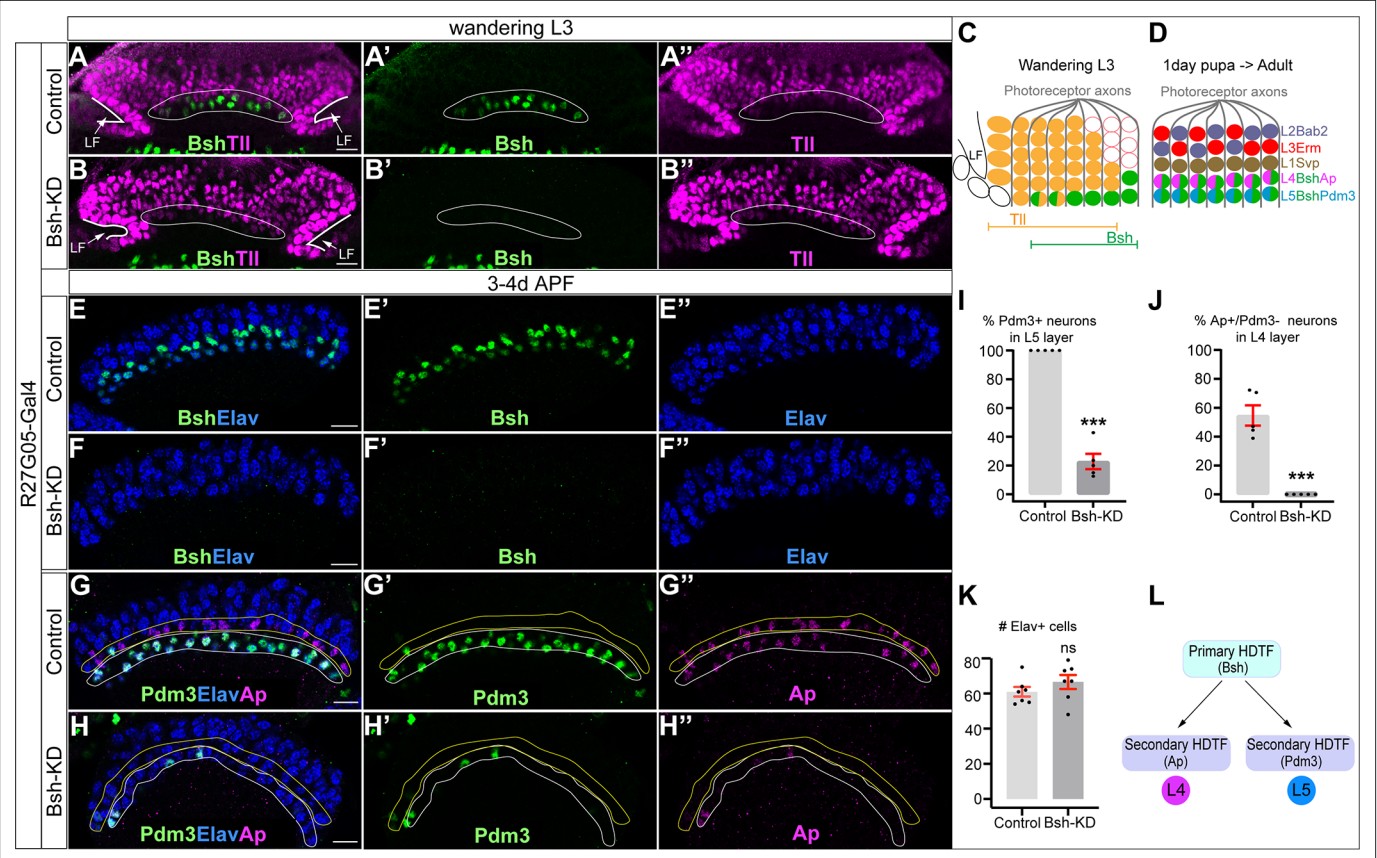

**Figure 2.** Bsh activates Ap/Pdm3 expression and specifies L4/L5 neuronal fate. (**A–B"**) Bsh-knockdown (KD) in lamina progenitor cells (LPCs) (R27G05-Gal4>UAS-Bsh-RNAi) eliminates Bsh in LPCs and neurons in wandering L3 (white circle). Tll labels all LPCs. LF: lamina furrow. L3: third larval instar. Here and below, scale bar: 10 μm, n≥5 brains. (**C**) Schematic of lamina neuron development at third larval instar. (**D**) Schematic of lamina neuron development from 1-day pupae to adult. (**E–F"**) Bsh remains undetectable in the lamina of 3- to 4-day pupa in Bsh-KD (R27G05-Gal4>UAS-Bsh-RNAi). (**G–K**) Bsh-KD in LPCs removes most L4 (Bsh/Ap) (**J**) and L5 (Bsh/Pdm3) (**I**) neuron markers. The Ap expression in L5 is caused by the Gal4 driver line but is irrelevant here. (**K**) The number of Elav+ cells in a single slice. n=5 brains in (**I**) and (**J**), n=7 brains in (**K**). L4 layer, yellow outline. L5 layer, white outline. (**L**) Summary. Data are presented as mean ± SEM. Each dot represents a brain. ***p<0.001, ns = not significant, unpaired t-test.

The online version of this article includes the following figure supplement(s) for figure 2:

**Figure supplement 1.** R27G05-GAL4 is turned on in all lamina progenitor cells (LPCs) and turned off in lamina neurons.

**Figure supplement 2.** Bsh-knockdown (KD) or Bsh-knockout (KO) in lamina progenitor cells (LPCs) affects L4/L5 neuronal fate.

**Figure supplement 3.** Bsh-knockdown (KD) in L4 neurons results in loss of detectable Bsh between 2 and 3 days after pupa formation (APF).

**Figure supplement 4.** Bsh is not required in L4 neurons to maintain L4 neuronal fate.

expression is unable to reinitiate in neurons if lost in LPCs (*Figure 2—figure supplement 2A'–B'*). Importantly, Bsh-KD resulted in a nearly complete loss of Ap and Pdm3 expression (*Figure 2G–J*); these neurons are not dying, as the number of Elav+ lamina neurons is unchanged (*Figure 2K*). Note that ectopic Ap expression in L5 is caused by the R27G05-Gal4 line, probably due to its genome insertion site, but this does not affect our conclusion that Bsh is required for Ap and Pdm3 expression. Consistent with the Bsh-KD phenotype, we found similar results using a Bsh Crispr/Cas9 knockout (KO) (*Figure 2—figure supplement 2E–I*). Taken together, we conclude that the primary HDTF Bsh is required to drive the expression of the secondary HDTFs Ap and Pdm3 and specify L4/L5 neuronal fates (*Figure 2L*).

To determine if Bsh is continuously required for Ap expression in L4, we first performed Bsh-KD specifically in L4 neurons using L4-split Gal4 (31C06-AD, 34G07-DBD)>UAS-Bsh-RNAi. Interestingly, this fails to knock down Bsh, suggesting L4-split Gal4 depends on Bsh expression. Next, we tried a Bsh Crispr/Cas9 KO (Bsh-KO: 31C06-AD, 34G07-DBD>UAS-Cas9, UAS-Bsh-sgRNAs) to remove Bsh expression beginning in L4 neurons, and it indeed led to a significant decrease in Bsh+ neurons

between 2 and 3 days APF (*Figure 2—figure supplement 3*). Despite the loss of Bsh expression in most L4 neurons, we observed no loss of Ap expression and no derepression of other lamina neuron markers (*Figure 2—figure supplement 4*). These results show (a) that all known lamina neuron markers are independent of Bsh regulation in neurons and (b) that Ap may undergo positive auto-regulation after its initiation, rendering it independent of Bsh. Autoregulation is a common feature of HDTFs (*Leyva-Díaz and Hobert, 2019*). We conclude that Bsh is not required to maintain Ap expression or repress other lamina neuron markers in L4 neurons.

## Bsh suppresses L1/L3 neuronal fates

We next asked whether L4/L5 neurons are transformed into another neuronal type following Bsh-KD (R27G05-Gal4>UAS-Bsh-RNAi). Above, we showed that the number of Elav+ lamina neurons remains unchanged in Bsh-KD, indicating that L4/L5 have assumed a different cell fate (*Figure 2K*). To test for ectopic generation of another lamina neuron type, we assayed the expression of Svp (L1), Bab2 (L2), and Erm (L3). We found that Bsh-KD led to ectopic expression of the L1 and L3 markers Svp and Erm in the positions normally occupied by L4/L5 cell bodies (*Figure 3A–E*). Notably, we never saw cell bodies co-expressing Erm and Svp, excluding the possibility of an ectopic hybrid neuronal fate in Bsh-KD. In contrast, the L2 marker Bab2 was unaffected by Bsh-KD (*Figure 3F–H*). Together, our data suggest that the absence of Bsh may generate ectopic L1 and L3 neuron types at the expense of L4 and L5.

To confirm L1 and L3 neurons are ectopically generated at the expense of L4 and L5 in Bsh-KD, we used Bsh-LexA>LexAop-GFP to trace L5 neurons (*Figure 3—figure supplement 1A–A''*). Indeed, control GFP+ neurons turn on L5 marker Pdm3 soon after their birth (*Figure 3I–I''*). In contrast, Bsh-KD generated GFP+ neurons that turn on L1 marker Svp instead of L5 marker Pdm3 (*Figure 3I–L*). Inter-estingly, some of the GFP+Svp+ neurons had cell bodies displaced from the L5 layer into the L1 layer (*Figure 3J*), which suggests that L1 neurons may actively seek out their appropriate settling layer. Furthermore, we also observed a transformation of L5 to L1 in neuronal morphology. In control, GFP+ neurons, which trace L5 neurons, have very few dendrites in the lamina neuropil (*Figure 3M*). In contrast, Bsh-KD resulted in GFP+ neurons elaborating L1-like dendrite arbors – bushy dendrites throughout the lamina (*Figure 3N and O*; summarized in *Figure 3P*). Because Bsh-KD generates L1 neuron type at the expense of L5, the ectopic L3 neurons must be generated at the expense of L4, which is confirmed by our accompanying work (*Xu et al., 2023*). This is also consistent with a previous report showing the transformation of L4 to L3 morphology in *bsh* mutant clones, although, unlike our results, they observed an L5 >glial fate change using L5-Gal4 to trace L5 in *bsh* clones (*Hasegawa et al., 2013*). The difference in results (where we see L5>L1 and they see L5>glia) is likely due to the unfaithful expression of L5-Gal4 in *bsh* mutants where L5 neuron type is missing. Together, we conclude that the absence of Bsh generates ectopic L1 and L3 neuron types at the expense of L5 and L4, respectively.

Bsh-KD in LPCs results in a loss of Ap expression and ectopic L1/L3 marker expression. To exclude the possibility that Bsh represses L1/L3 fates through Ap, we knocked down Ap expression from their time of birth using an LPC-Gal4 line (R27G05-Gal4>Ap-RNAi). As expected, Ap-KD eliminates Ap expression in L4 neurons (*Figure 3—figure supplement 2*). Furthermore, Ap remained undetectable in lamina neurons in the adult, which indicates that Ap expression is unlikely able to reinitiate if normal initiation is lost (*Figure 3—figure supplement 2*). Importantly, loss of Ap did not affect Bsh expression in L4 and did not lead to ectopic expression of other lamina neuron markers (*Figure 3—figure supplement 3*), suggesting that Bsh but not Ap is required to repress L1/L3 neuronal fates. Taken together, we conclude that the primary HDTF Bsh, but not the secondary HDTF Ap, promotes L4/L5 neuronal fates and suppresses L1/L3 fates, thereby generating lamina neuronal diversity (*Figure 3P*).

## Bsh represses Zfh1 to suppress L1/L3 neuronal fates

How does the primary HDTF Bsh repress L1 and L3 neuronal fates? We hypothesize that Bsh might repress an unidentified HDTF, which is shared by L1 and L3 and required to generate L1 and L3 neuron types. To find this HDTF, we screened published RNA-seq data (*Tan et al., 2015*) and found that the HDTF Zfh1 was present in all LPCs before becoming restricted to L1 and L3 neurons (*Figure 4A–D*). To determine if Zfh1 is required for L1 and L3 neuronal fates, we used RNAi to perform Zfh1-KD in LPCs (R27G05-Gal4>Zfh1-RNAi). As expected, Zfh1-KD significantly decreased Zfh1 nuclear levels in all LPCs and neurons (*Figure 4D–E''*; *Figure 4—figure supplement 1A–D*) and resulted in a loss of Svp+

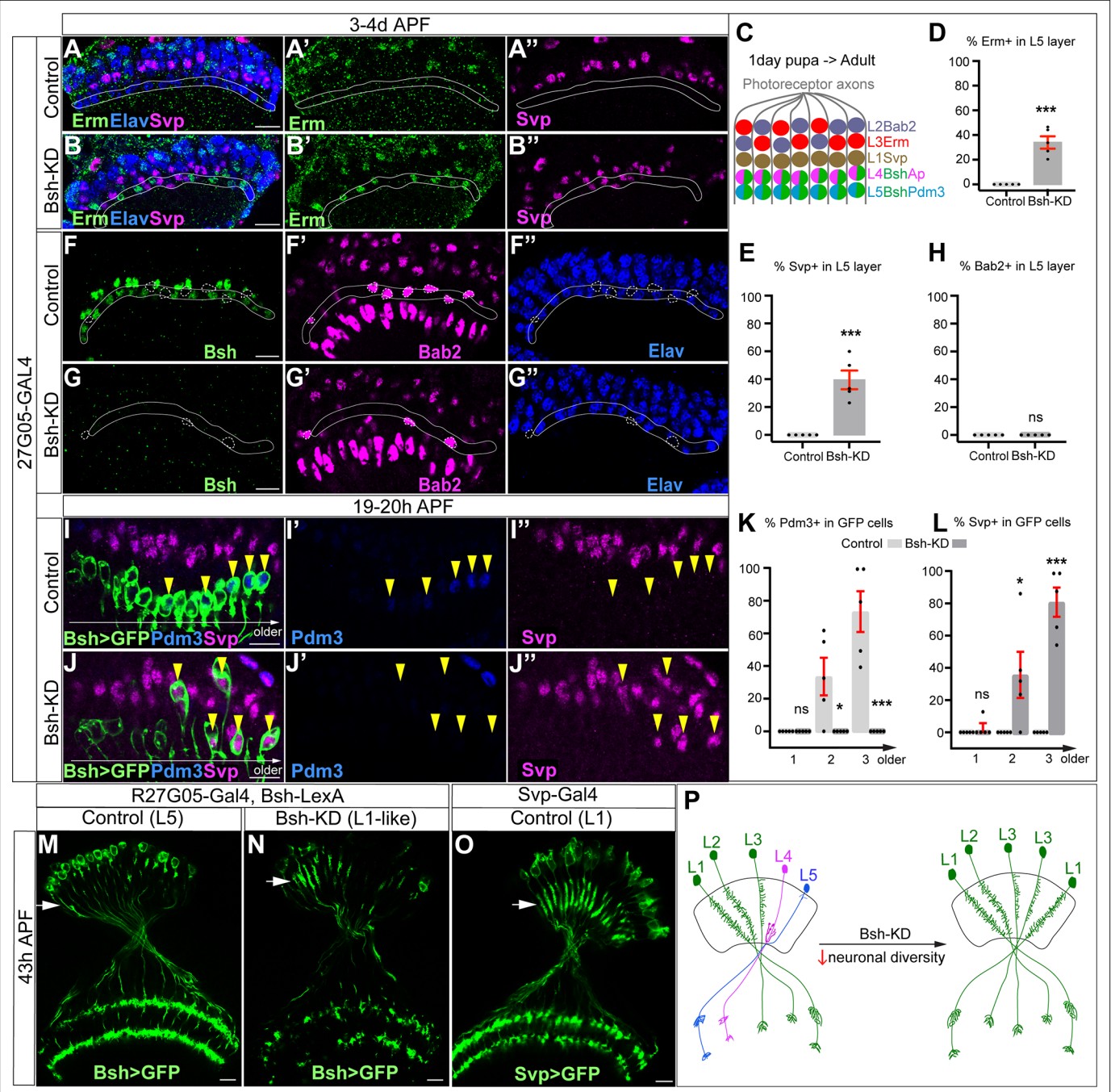

**Figure 3.** Bsh suppresses L1/L3 neuronal fate. (A–E) Bsh-knockdown (KD) in lamina progenitor cells (LPCs) results in the ectopic expression of the L1 marker Svp and L3 marker Erm in L4/L5 cell body layers (circled). (C) Schematic of lamina neuron development from 1-day pupae to adult. (D and E) Quantification of Erm and Svp expression. Here and below, scale bar, 10 μm. (F–H) Bsh-KD in LPCs does not produce ectopic Bab2-positive neurons or glia in the L5 layer (circled). n≥5 brains. Genotype: R27G05-Gal4>UAS-Bsh-RNAi. (H) Quantification of Bab2 expression. (I–L) Bsh-KD in LPCs results in ectopic Svp+ L1 neurons at the expense of Pdm3+ L5 neurons. Bsh GFP+ neurons marked with yellow arrowheads show L1 marker Svp expression in Bsh-KD while L5 marker Pdm3 expression is in control. Genotype: Bsh-LexA>LexAop-GFP. (K and L) Quantification of Pdm3 and Svp expression. (M–O) Bsh-KD transforms L5 neuron morphology to L1-like neuronal morphology. (M) Control L5 neurons have very few dendrites in the lamina neuropil. Genotype: R27G05-Gal4, Bsh-LexA>LexAop-GFP. (N) Bsh-KD transforms L5 neuron morphology to L1-like neuronal morphology. Genotype: R27G05-Gal4>UAS-Bsh-RNAi; Bsh-LexA>LexAop-GFP. (O) Control L1 neurons show bushy dendrites throughout the lamina. Genotype: svp-Gal4, R27G05-FLP>UAS-FRT-stop-FRT-myrGFP. (P) Summary. Data are presented as mean ± SEM. Each dot represents each brain. n=5 brains in (D), (E), (H), (K), and (L). *p<0.05, **p<0.01, ***p<0.001, ns = not significant, unpaired t-test.

The online version of this article includes the following figure supplement(s) for figure 3:

*Figure 3 continued on next page*

*Figure 3 continued*

**Figure supplement 1.** Loss of Bsh in lamina progenitor cells (LPCs) transforms L5 neurons into L1 neurons.

**Figure supplement 2.** Ap-knockdown (KD) eliminates Ap initiation and maintenance in L4 neurons.

**Figure supplement 3.** Ap is not required to suppress other lamina neuron markers in L4 neurons.

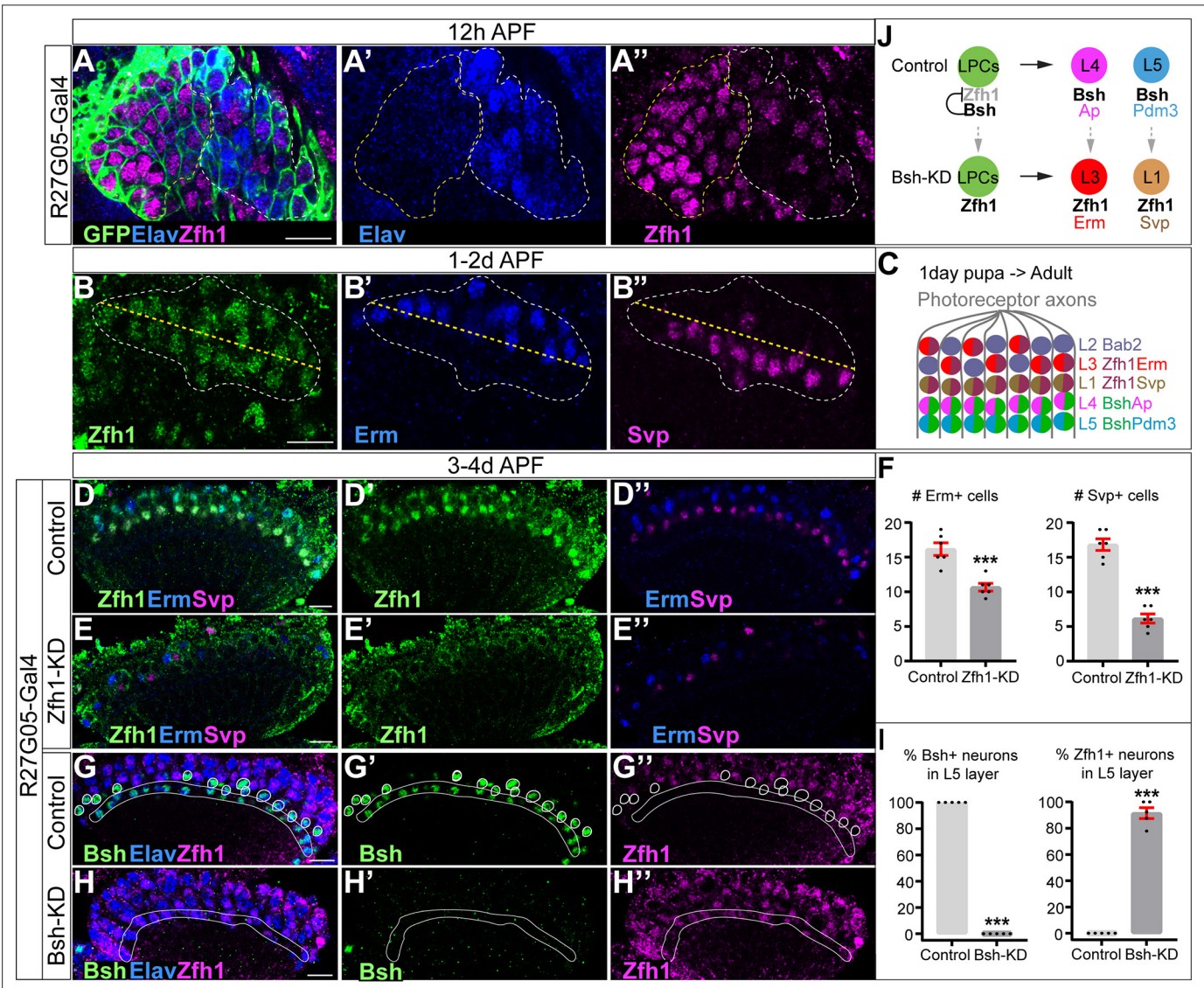

**Figure 4.** Bsh represses Zfh1 to suppress L1/L3 neuronal fate. (**A–A″**) Zfh1 is expressed in all lamina progenitor cells (LPCs) and some lamina neurons at 12 hr after pupa formation (APF). GFP labels all lamina cells. Elav labels lamina neurons. The yellow dash circle outlines LPCs and the white dash circle outlines lamina neurons. Genotype: R27G05-Gal4>UAS-myrGFP. Here and below, scale bar, 10 μm. n≥5 brains. (**B–B″**) Zfh1 is expressed in Svp+ L1 and Erm+ L3 neurons at 1–2 days APF; Svp and Erm are never co-expressed. The white dashed circle outlines L1 and L3 neurons and the yellow line indicates the rough boundary between L1 and L3 cell bodies. (**C**) Schematic of lamina neuron development from 1-day pupae to adult. (**D–F**) Zfh1-knockdown (KD) in LPCs results in a loss of Svp+ L1 and Erm+ L3 neurons. Quantification: the number of Erm+ or Svp+ cell bodies in a single optical slice. Genotype: R27G05-Gal4>UAS-zfh1-RNAi. (**G–I**) Bsh-KD in LPCs results in ectopic Zfh1 in L4/L5 layers. White circles label Bsh+ cell bodies in L4 layer in control. L5 layer, white outline. Quantification: the percentage of Bsh+ or Zfh1+ neurons in L5 layer. Genotype: R27G05-Gal4>UAS-bsh-RNAi. (**J**) Summary. Data are presented as mean ± SEM. Each dot represents each brain. n=6 brains in (**F**) and n=5 brains in (**I**). ***p<0.001, unpaired t-test.

The online version of this article includes the following figure supplement(s) for figure 4:

**Figure supplement 1.** Zfh1 expression and epistasis with Bsh.

L1 and Erm+ L3 neurons (*Figure 4D–F*), consistent with a role in specifying L1 and L3 neuronal identity. In contrast, Zhf1-KD did not increase the total number of Ap/Pdm3+ neurons (*Figure 4—figure supplement 1E–G*), showing that Zfh1 does not repress L4/L5 neuronal identity. We observed that Zfh1-KD resulted in fewer lamina neurons overall, including a reduction of Bsh+ L4 and L5 neurons (*Figure 4—figure supplement 1E'–K*); this suggests a potential role for Zfh1 in LPCs in regulating lamina neurogenesis. Importantly, Bsh-KD (R27G05-Gal4>Bsh-RNAi) resulted in ectopic expression of Zfh1 in the L4/L5 cell body layers and in GFP+ neurons (normally L5) (*Figure 4G–I*, *Figure 4—figure supplement 1L–N*). We propose that Bsh and Zfh1 are both primary HDTFs (specifying L4/L5 and L1/L3, respectively) and that Bsh represses Zfh1 to suppress L1/L3 neuronal fates (*Figure 4J*).

## Bsh:DamID reveals Bsh direct binding to L4 identity genes and pan-neuronal genes

In *C. elegans,* terminal selectors show non-redundant control of neuronal identity genes and redundant control of pan-neuronal genes. To see if the same mechanism is used by the HDTF Bsh, we profiled Bsh direct targets with precise spatial and temporal control: only in L4 neurons at the time of synapse formation (46–76 hr APF). To do this, we used Targeted DamID (*Aughey et al., 2021*), which can be used to identify Bsh DNA-binding sites across the genome (*Figure 5A*). We generated a Bsh:Dam transgenic fly line according to published methods (*Aughey et al., 2021*) and expressed it specifically in L4 neurons using the L4-Gal4 transgene R31C06-Gal4 during synapse formation (*Figure 5—figure supplement 1A–B""*). We verified that the Bsh:Dam fusion protein was functional by rescuing Ap and Pdm3 expression (*Figure 5—figure supplement 1C–E""*). We performed three biological replicates which had high reproducibility (*Figure 5B*).

Next, we determined which Bsh-bound genomic targets showed enriched transcription in L4 neurons during synapse formation using recently published L4 scRNA sequencing data from the same stage (GEO: GSE190714) (*Jain et al., 2022*). There are 958 genes that are significantly transcribed in L4 at 48 hr or 60 hr APF. Among them, 421 genes show Bsh:Dam binding peaks, while 537 genes do not show Bsh:Dam binding peaks (*Figure 5C*). Genes having Bsh:Dam binding peaks include numerous candidate L4 identity genes: ion channels, synaptic organizers, cytoskeleton regulators, synaptic recognition molecules, neuropeptide/receptor, neurotransmitter/receptor, and pan-neuronal genes (*Figure 5DSupplementary file 1*). Genes expressed in L4 but not having Bsh:Dam binding peaks include long non-coding RNA, mitochondrial genes, ribosomal protein, heat shock protein, ATP synthase, and others (*Supplementary file 2*).

We previously showed that DIP-β, a cell surface protein of the immunoglobulin superfamily, is specifically expressed in L4 neurites in the proximal lamina, and is required for proper L4 circuit formation (*Xu et al., 2019*). Here, we found Bsh binding peaks in L4 neurons within the first intron of the *DIP-β* gene (*Figure 5D*). Interestingly, the Ecdysone receptor (EcR), which controls the temporal expression of *DIP-β* in L4 neurons, also has a DNA-binding motif in the *DIP-β* first intron (*Jain et al., 2022*), suggesting Bsh and the EcR pathway may cooperate to achieve the proper spatial (Bsh) and temporal (EcR) expression pattern of the DIP-β synapse recognition molecule. Taken together, consistent with the work of terminal selector in *C. elegans,* we found evidence that Bsh:Dam shows direct binding to L4 identity genes – including *DIP-β* – as well as pan-neuronal genes (summarized in *Figure 5E*).

Although our experiment focused on Bsh targets during the stages of synapse formation, we checked for Bsh binding at the HDTF loci that might be bound by Bsh during the earlier stages of neuronal identity specification. We found that Bsh:Dam showed a Bsh binding peak at the *ap* locus, suggesting that Bsh may redundantly maintain Ap expression in L4 neurons through the stage of synapse formation (along with potential Ap-positive autoregulation) (*Figure 5—figure supplement 1F*). In contrast, Bsh:Dam did not show binding peaks at *pdm3* or *zfh1* loci, which might be due to the inaccessibility of *pdm3* or *zfh1* loci in L4 neurons. Indeed, Dam (open chromatin) (*Aughey et al., 2018*) does not show a peak at *pdm3* or *zfh1* loci, suggesting that *pdm3* or *zfh1* loci are not accessible in L4 neurons during synaptogenesis (*Figure 5—figure supplement 1F*).

## Bsh and Ap form a coherent feed-forward loop to activate DIP-β

Here, we ask whether Bsh or Ap are required for the expression of DIP-β in L4 neurons. We found that Bsh-KO (31C06-AD, 34G07-DBD>UAS-Cas9, UAS-Bsh-sgRNAs) only in L4 neurons resulted in a strong decrease in DIP-β levels (*Figure 6A–D*). Furthermore, using the STaR method (*Chen et al.,*

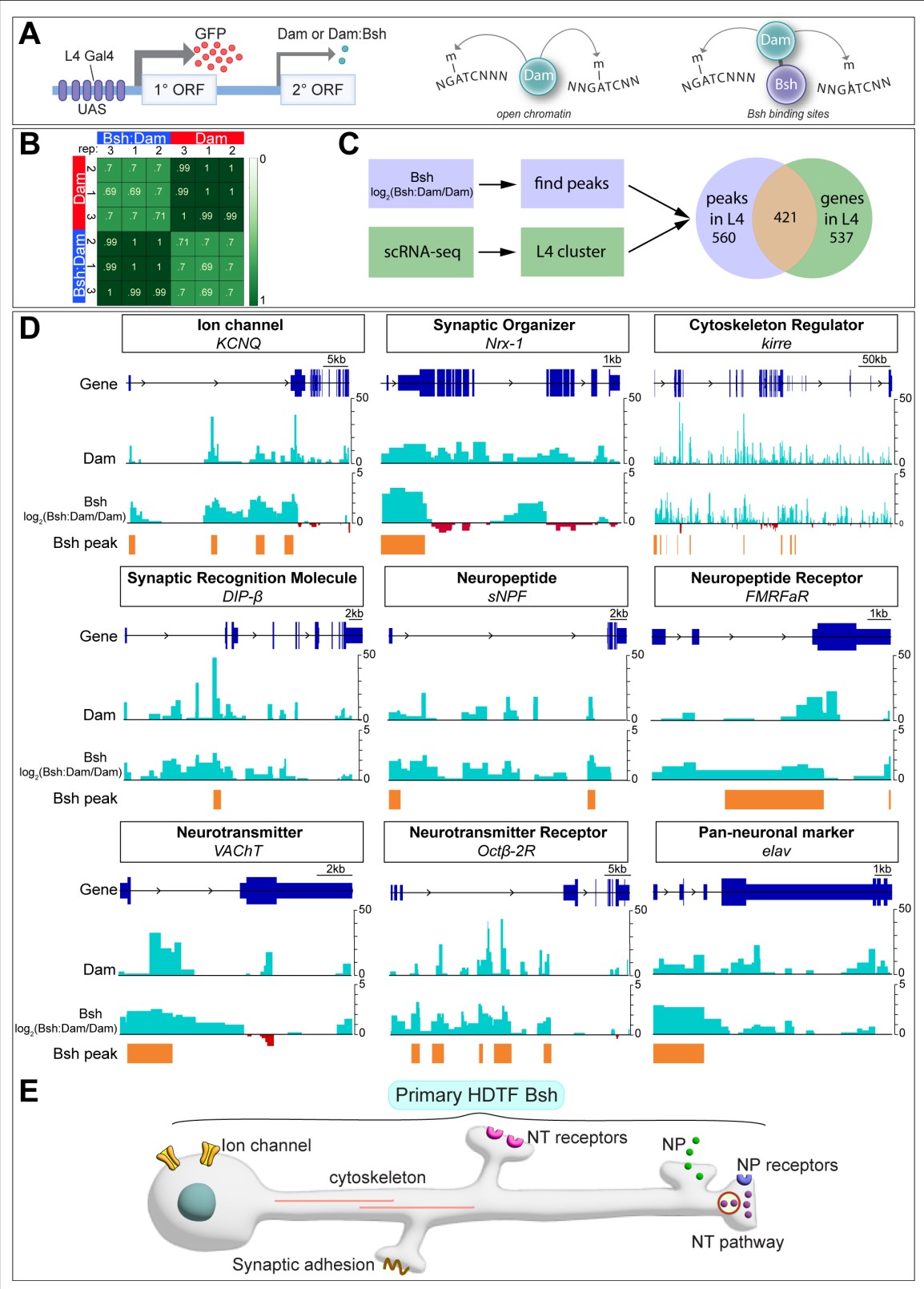

**Figure 5.** Bsh:DamID reveals Bsh direct binding to L4 identity genes and pan-neuronal genes. (**A**) Schematic for TaDa method. (**B**) Both Dam and Bsh:Dam show high Pearson correlation coefficients among their three biological replicates, with lower Pearson correlation coefficients between Dam and Bsh:Dam. A heatmap is generated using the multiBamSummary and plotCorrelation functions of deepTools. (**C**) Bsh:Dam peaks in L4 (46–76 hr after pupa formation [APF]) called by find_peaks and L4 scRNAseq data (48 hr APF; 60 hr APF) (*Jain et al., 2022*) are combined (see Materials and

*Figure 5 continued on next page*

*Figure 5 continued*

methods). (**D**) Bsh:Dam shows strong signals in L4 identity genes and pan-neuronal genes. The Dam alone signal indicates the open chromatin region in L4 neurons. The y axes of Bsh:Dam signal represent log2 (Bsh:Dam/Dam) scores. Bsh peaks in L4 neurons were generated using find_peaks (FDR<0.01; min_quant = 0.9) and peaks2genes. (**E**) Summary.

The online version of this article includes the following figure supplement(s) for figure 5:

**Figure supplement 1.** UAS-Bsh:Dam expressed by 31C06-Gal4 is functional and specifically expressed in L4 neurons; Bsh:Dam shows a binding peak in *ap* locus but not *pdm3* or *zfh1* in L4 neurons.

*2014*; *Xu et al., 2019*), we found that L4 primary dendrite length and presynaptic Bruchpilot (Brp) puncta in the proximal lamina were both decreased following Bsh-KO (*Figure 6E–I*); this is distinct from the DIP-β-KD phenotype (*Xu et al., 2019*). We conclude that Bsh is required to activate DIP-β expression in L4 and regulate L4 morphology and connectivity. Because Bsh-KO has a distinct phenotype from DIP-β-KD, it is likely that Bsh has distinct targets in addition to DIP-β.

To test whether Ap controls DIP-β expression in L4, we knocked down Ap in L4 neurons (R27G05-Gal4>Ap-RNAi) and observed a strong decrease in DIP-β levels in L4 neurons (*Figure 6J–M*). Note that L4-split Gal4 (31C06-AD, 34G07-DBD)>UAS-Ap-RNAi cannot knock down Ap in L4 neurons due to the dependence of L4-split Gal4 on Ap expression. We next combined the STaR method with Ap-shRNA (see Materials and methods), resulting in a loss of Ap expression in L4 at 2 days APF and a strong decrease in DIP-β levels in L4 neurons (*Figure 6—figure supplement 1*). Importantly, Ap-KD in L4 neurons increased Brp puncta in the distal/proximal lamina and increased the length of L4 primary dendrites, which is similar to DIP-β-KD phenotype (*Xu et al., 2019*; *Figure 6N–R*). We conclude that Ap is required to activate DIP-β expression in L4 and regulate L4 morphology and connectivity mainly through DIP-β.

A coherent feed-forward motif is A activates B, followed by A and B, both activating C (*Mangan and Alon, 2003*). This is what we observe for Bsh, Ap, and DIP-β. Bsh activates Ap in L4 neurons soon after their birth (*Figure 2G and H*), and Bsh and Ap are both required to activate DIP-β (*Figure 6A–D and J–M*). Importantly, loss of Bsh in L4 neurons decreases DIP-β levels (*Figure 6A–D*) without altering Ap expression (*Figure 2—figure supplement 4F*). Similarly, loss of Ap has no effect on Bsh (*Figure 3—figure supplement 3A–B'''*), yet it decreases DIP-β levels (*Figure 6J–M'''*). In conclusion, we have defined a coherent feed-forward loop in which Bsh activates Ap, and then both are independently required to promote expression of the synapse recognition gene DIP-β, thereby bridging neuronal fate decision to synaptic connectivity (*Figure 6*).

## Bsh is required for normal visual behavior

Bsh is required to specify L4/L5 neuronal fate and generate lamina neuronal diversity (control: L1-L5; Bsh-KD: L1-L3) (*Figure 3*), raising the hypothesis that lack of Bsh may compromise lamina function. To test this, we used an apparatus (the Fly Vision Box) that integrates multiple assays, including visual motion (*Zhu et al., 2009*), phototaxis (*Benzer, 1967*), and spectral preference (*Gao et al., 2008*) in flies walking in transparent tubes (*Isaacson et al., 2023*; *Figure 7*; see Materials and methods). We used the LPC-specific driver R27G05-Gal4 to express Bsh RNAi in LPCs. We found that Bsh-KD in LPCs resulted in a lack of Bsh in the adult lamina (*Isaacson et al., 2023*; *Figure 7—figure supplement 1*). Flies with Bsh-KD in LPCs, where L4/L5 neurons are transformed into L1/L3 neurons, showed a reduced response to a high-speed stimulus, suggesting weakened sensitivity to visual motion (*Figure 7B*). Previous work found that L4 function is not required for motion detection when silencing L4 or L5 neuron activity alone (*Silies et al., 2013*; *Tuthill et al., 2013*), which suggests that L4 and L5 acting together might be required for normal sensitivity to visual motion. When tested with both UV and green LEDs, the Bsh-KD flies had reduced phototaxis to both dim and bright lights, suggesting less sensitivity to both UV and green lights. Interestingly, the Bsh-KD flies exhibited larger responses toward bright UV illumination in the spectral preference assay (*Figure 7C and D*). This apparent attraction to bright UV light may result from more weakened sensitivity to green light, which may be expected since L4 and L5 are indirect R1-6 targets (*Takemura et al., 2015*). Taken together, we conclude that the primary HDTF Bsh is required to generate lamina neuronal diversity and normal visual behavior (*Figure 7E*).

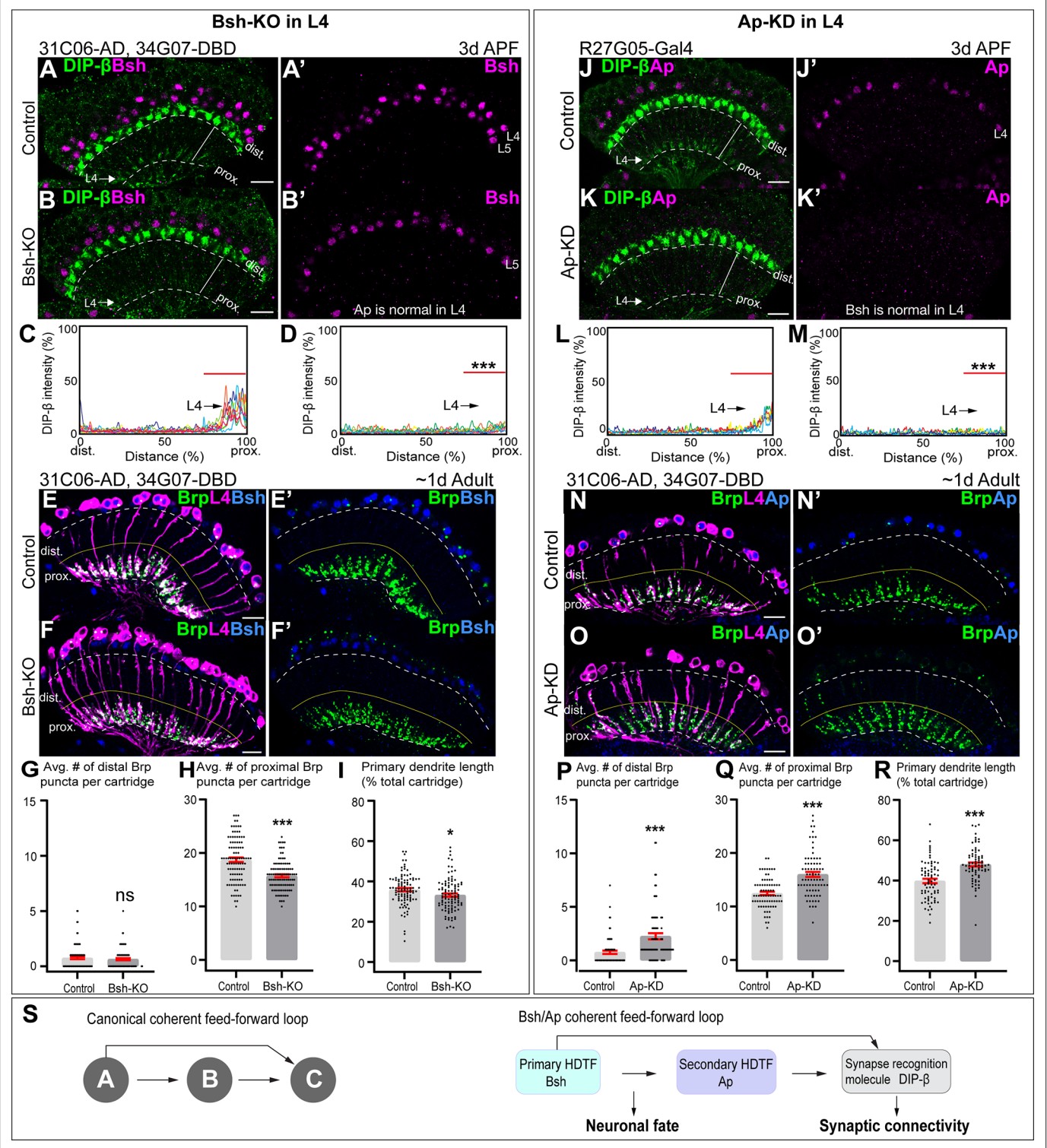

**Figure 6.** Bsh and Ap form a coherent feed-forward loop to activate DIP-β. (**A–D**) Bsh Crispr knockout (KO) in postmitotic L4 neurons results in loss of DIP-β expression in the proximal lamina neuropil (arrow) at 3 days after pupa formation (APF). DIP-β expression is detected using an anti-DIP-β antibody. The signal in the distal lamina is from non-lamina neurons, probably LawF. Significantly reduced DIP-β fluorescence intensity is observed in the proximal lamina (75–100% distance, marked by red bar (**C, D**)). ***p<0.001, unpaired t-test, n=8 brains, each line represents each brain, scale bar, 10 µm. Genotype: 31C06-AD, 34G07-DBD>UAS-Cas9, UAS-Bsh-sgRNAs. (**E–I**) Bsh Crispr KO in L4 neurons results in a decrease of primary dendrite length and proximal synapse number in postmitotic L4 neurons of 1-day adults. Here and below, white dash lines indicate the lamina neuropil and yellow lines show the boundary between the distal and proximal lamina. The average number of Brp puncta in L4 neurons present within the distal or proximal

*Figure 6 continued on next page*

*Figure 6 continued*

halves of lamina cartridges. *p<0.05, ***p<0.001, ns = not significant, unpaired t-test, n=100 cartridges, n=5 brains, each dot represents one cartridge, data are presented as mean ± SEM. Genotype: 31C06-AD, 34G07-DBD>UAS-Cas9, UAS-Bsh-sgRNAs, UAS-myrGFP, UAS-RSR, 79C23-S-GS-rst-stop-rst-smFPV5-2a-GAL4. (**J–M**) Ap RNAi knockdown (KD) in postmitotic L4 neurons results in loss of DIP-β expression in the proximal lamina neuropil (arrow) at 3 days APF. The signal in the distal lamina is from non-lamina neurons, probably LawF. Significantly reduced DIP-β fluorescence intensity is observed in the proximal lamina (75%–100% distance, marked by red bar (**L, M**)). ***p<0.001, unpaired t-test, n=8 brains, each line represents each brain, scale bar, 10 μm. Genotype: R27G05-Gal4>UAS-ApRNAi. (**N–R**) Ap-KD in L4 neurons results in an increase of primary dendrite length and proximal synapse number in postmitotic L4 neurons in 1-day adults. The average number of Brp puncta in L4 neurons present within the distal or proximal halves of lamina cartridges. *p<0.05, ***p<0.001, ns = not significant, unpaired t-test, n=100 cartridges, n=5 brains, each dot represents one cartridge, data are presented as mean ± SEM. Genotype: 31C06-AD, 34G07-DBD>UAS-RSR, 79C23-S-GS-rst-stop-rst-smFPV5-2a-GAL4, UAS-Ap-shRNA, UAS-myrGFP. (**S**) Summary.

The online version of this article includes the following figure supplement(s) for figure 6:

**Figure supplement 1.** DIP-β expression is disrupted when knocking down Ap in L4 neurons.

## Discussion

HDTFs are evolutionarily conserved factors in specifying neuron-type-specific structure and function (*Hobert, 2021*; *Hobert and Kratsios, 2019*; *Kitt et al., 2022*). In *C. elegans*, some HDTFs function as terminal selectors, controlling the expression of all neuronal identity genes and diversifying neuronal subtypes, while other HDTFs act downstream of terminal selectors to activate a subset of identity genes (*Gordon and Hobert, 2015*; *Hobert, 2016*). Here, we show that the Bsh

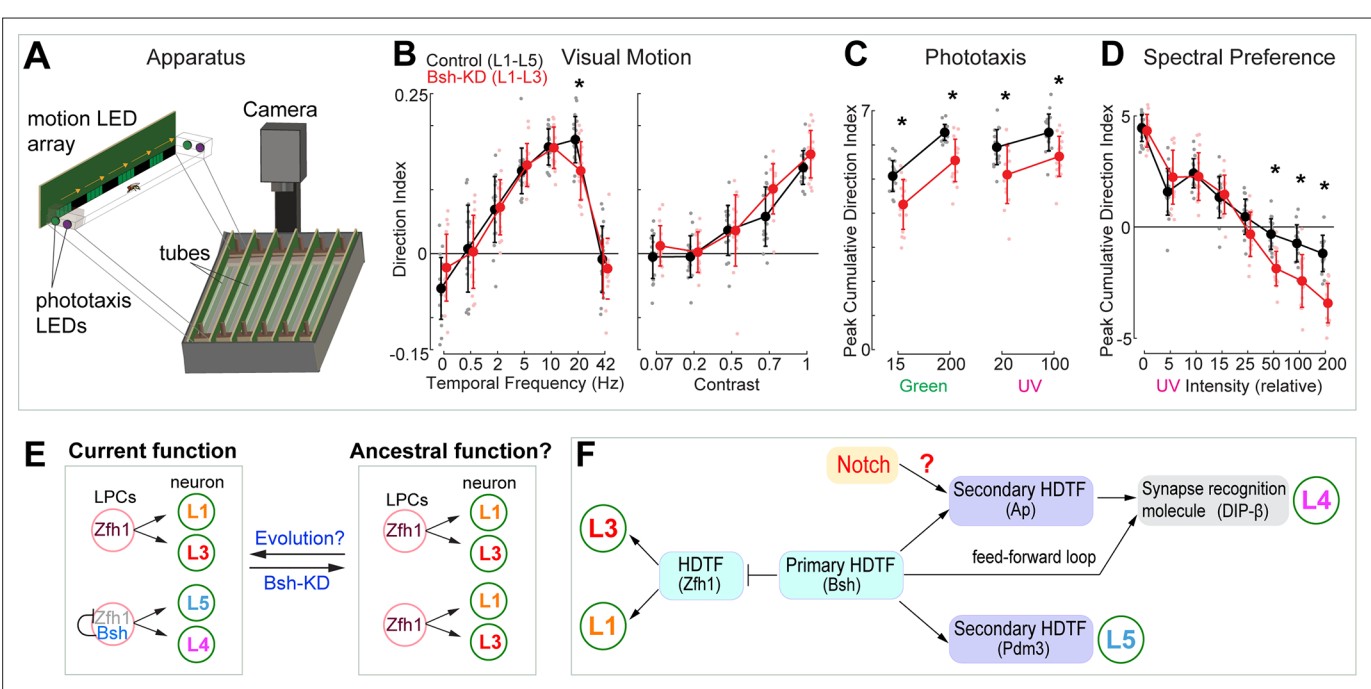

**Figure 7.** Bsh+ L4/L5 are required for normal visual sensitivity. (**A**) Schematic of the Fly Vision Box. (**B**) Bsh-knockdown (KD) adult flies show reduced responses to a high-speed stimulus. Left: stimulus with different temporal frequency; right: stimulus with 5 Hz temporal frequency, but with the indicated contrast level. (**C**) Bsh-KD adult flies show reduced phototaxis to both dim and bright lights. Relative intensity was used: 15 and 200 for dim and bright green, respectively; 20 and 100 for dim and bright UV, respectively. (**D**) Bsh-KD adult flies show larger responses toward bright UV illumination. For lower UV levels, flies walk toward the green LED, but walk toward the UV LED at higher UV levels. Data are presented as mean ± SD, with individual data points representing the mean value of all flies in each tube. For control (mherry-RNAi), n=18 groups of flies (tubes), for Bsh-KD, n=16 groups of flies, run across three different experiments. Each group is 11–13 male flies. *p<0.05, unpaired, two-sample t-test controlled for false discovery rate. (**E**) Model. Left: In wild type, Zfh1+ lamina progenitor cells (LPCs) give rise to L1 and L3 neurons, whereas Zfh1+Bsh+ LPCs give rise to L4 and L5 neurons. The lineage relationship between these neurons is unknown. Right: Bsh KD results in a transformation of L4/L5 into L1/L3 which may reveal a simpler, ancestral pattern of lamina neurons that contains the core visual system processing arrangement. (**F**) Summary.

The online version of this article includes the following figure supplement(s) for figure 7:

**Figure supplement 1.** Bsh-knockdown (KD) in lamina progenitor cells (LPCs) results in loss of Bsh in the adult fly.

primary HDTF functions for L4/L5 fate specification by promoting expression of the Ap and Pdm3 secondary HDTFs and suppressing the HDTF Zfh1 to inhibit ectopic L1/L3 fate, thereby generating lamina neuronal diversity. In L4, Bsh and Ap act in a feed-forward loop to drive the expression of synapse recognition molecule DIP-β, thereby bridging neuronal fate decision to synaptic connectivity (*Figure 7F*). Our DamID data provides support for several hundred Bsh direct binding targets that also show enriched expression in L4 neurons; these Bsh targets include predicted and known L4 identity genes as well as pan-neuronal genes, similar to the regulatory logic first observed in *C. elegans* (*Hobert, 2021*; *Stefanakis et al., 2015*; *Figure 5*). HDTFs are widely expressed in the nervous system in flies, worms, and mammals. By characterizing primary and secondary HDTFs according to their initiation order, we may decode conserved mechanisms for generating diverse neuron types with precise circuits assembly.

How can a single primary HDTF Bsh activate two different secondary HDTFs and specify two distinct neuron fates: L4 and L5 (*Figure 7F*)? In our accompanying work (*Xu et al., 2023*), we show that Notch signaling is activated in newborn L4 but not in L5. This is not due to an asymmetric partition of a Notch pathway component between sister neurons, as is common in most regions of the brain (*Li et al., 2013*; *Mark et al., 2021*), but rather due to L4 being exposed to Delta ligand in the adjacent L1 neurons; L5 is not in contact with the Delta+ L1 neurons and thus does not have active Notch signaling. We show that while Notch signaling and Bsh expression are mutually independent, Notch is necessary and sufficient for Bsh to specify L4 fate over L5. The Notch$^{ON}$ L4, compared to Notch$^{OFF}$ L5, has a distinct open chromatin landscape which allows Bsh to bind distinct genomic loci, leading to L4-specific identity gene transcription. We propose that Notch signaling and HDTF function are integrated to diversify neuronal types.

We used DamID (this work) and an scRNAseq dataset (*Jain et al., 2022*) to identify genomic loci containing both Bsh direct binding sites and L4-enriched expression. Genes that have Bsh:Dam binding peaks but are not detected in L4 scRNAseq data at 48 hr or 60 hr APF might be due to the following reasons: they are transcribed later, at 60–76 hr APF; the algorithm (find_peaks; peaks2genes) that we used to detect Bsh:Dam peaks and call the corresponding genes is not 100% accurate; some regulatory regions are outside the stringent ±1 kb association with genes; Bsh may act as transcription repressor; TFs generally act combinatorially as opposed to alone and that many required specific cooperative partner TFs to also be bound at an enhancer for gene activation; and scRNAseq data is not 100% accurate for representing gene transcription (*Figure 5C*; *Supplementary file 3*).

Does the primary HDTF Bsh control all L4 neuronal identity genes? It seems likely, as Bsh:Dam shows binding to L4-transcribed genes that could regulate L4 neuronal structure and function, including the functionally validated synapse recognition molecule DIP-β. Furthermore, we found Bsh and Ap form a feed-forward loop to control DIP-β expression in L4 neurons. Similarly, in *C. elegans*, terminal selectors UNC-86 and PAG-3 form a feed-forward loop with HDTF CEH-14 to control the expression of neuropeptide FLP-10, NLP-1, and NLP-15 in BDU neurons (*Gordon and Hobert, 2015*), suggesting an evolutionarily conserved approach, using feed-forward loops, for terminal selectors to activate neuronal identity genes. An important future direction would be testing whether Bsh controls the expression of all L4 identity genes via acting with Ap in a feed-forward loop. One intriguing approach would be profiling the Ap genome-binding targets in L4 during the synapse formation window and characterizing the unique and sharing genome-binding targets of Bsh and Ap in L4 neurons. Further, it would be interesting to test whether the primary HDTF Bsh functions with Ap to maintain neuron-type-specific morphology, connectivity, and function properties in adults.

Newborn neurons are molecularly distinct prior to establishing their characteristic morphological or functional attributes. We discovered that the primary HDTF Bsh is specifically expressed in newborn L4 and L5 neurons and is required to specify L4 and L5 fates, suggesting that identifying differentially expressed factors in newborn neurons is essential to decoding neuron-type specification. We note that primary and secondary TFs may be HDTFs as well as non-HDTFs. For example, the primary HDTF Zfh1 is required to activate Svp in L1 and Erm in L3, neither of which are HDTF, though Erm has a significant function in L3 axon targeting (*Peng et al., 2018*). This suggests that the primary HDTF can activate non-HDTFs to initiate neuron identity features. Recent work in *Drosophila* medulla found that a unique combination of TFs (a mix of HDTFs and non-HDTFs) is required to control neuron identity features (*Özel et al., 2022*). It would be important to dissect whether there is hierarchical expression and function within these TF combinations and to test whether HDTFs activate non-HDTFs.

Evolution can drive a coordinated increase in neuronal diversity and functional complexity. We hypothesize that there was an evolutionary path promoting increased neuronal diversity by the addition of primary HDTF Bsh expression. This is based on our finding that the loss of a single HDTF (Bsh) results in reduced lamina neuron diversity (only L1-L3), which may represent a simpler ancestral brain. A similar observation was described in *C. elegans* where the loss of a single terminal selector caused two different neuron types to become identical, which was speculated to be the ancestral ground state (*Arlotta and Hobert, 2015*; *Cros and Hobert, 2022*; *Reilly et al., 2022*), suggesting phylogenetically conserved principles observed in highly distinct species. An interesting possibility is that evolutionarily primitive insects, such as silverfish (*Truman and Riddiford, 1999*), lack Bsh expression and L4/L5 neurons, retaining only the core motion detection L1-L3 neurons. Our findings provide a testable model whereby neural circuits evolve more complexity by adding the expression of a primary HDTF (*Figure 7E*).

# Materials and methods
## Contact for reagent and resource sharing
Further information and requests for resources and reagents should be directed to and will be fulfilled by the Lead Contact Chundi Xu (cxu3@uoregon.edu) and Chris Doe (cdoe@uoregon.edu).

## Experimental model and subject details
All flies were reared at 25°C on standard cornmeal fly food, unless otherwise stated. For all RNAi and shRNA KD experiments, crosses are kept at 25°C and their progeny are kept at 28.5°C with 16:8 hr light-dark cycle from the embryo stage until dissection. For all Gal80ts experiments, crosses are kept at 18°C and progenies are kept at 29°C at the desired time.

## Method details
### Animal collections
For the Bsh-misexpression experiment, crosses were reared at 25°C in collection bottles fitted with 3.0% agar apple juice caps containing plain yeast paste. Embryos were then collected on 3.0% agar apple juice caps with plain yeast paste for 4 hr. The collected embryos were moved to 18°C until 72 hr after larval hatching (ALH). The larvae at 72 hr ALH were moved to 29°C until 58 hr after pupal formation.

For the experiment R27G05-GAL4>UAS-myrGFP, tubP-GAL80[ts], the progeny is kept in 18°C from embryo and moved to 29.2°C at early wandering L3 or 1 day APF for 20 hr.

For the behavioral experiments, the progeny is kept in 18°C from embryo and moved to 29°C with 16:8 hr light-dark cycle from the early larval stage until behavioral tests. Male flies at 2–5 days after the eclosion at 29°C were used for the Fly Vision Box experiments.

### Immunohistochemistry
Fly brains were dissected in Schneider's medium and fixed in 4% paraformaldehyde in phosphate buffered saline (PBS) for 25 min. After fixation, brains were quickly washed with PBS with 0.5% Triton X-100 (PBT) and incubated in PBT for at least 2 hr at room temperature. Next, samples were incubated in blocking buffer (10% normal donkey serum, 0.5% Triton X-100 in PBS) overnight at 4°C. Brains were then incubated in primary antibody (diluted in blocking buffer) at 4°C for at least two nights. Following primary antibody incubation, brains were washed with PBT. Next, brains were incubated in secondary antibody (diluted in blocking buffer) at 4°C for at least 1 day. Following secondary antibody incubation, brains were washed with PBT. Finally, brains were mounted in SlowFade Gold antifade reagent (Thermo Fisher Scientific, Waltham, MA, USA).

Images were acquired using a Zeiss 800 confocal and processed with ImageJ and Adobe Photoshop.

### Knocking down HDTF in neurons
Genotype for knocking down Ap specifically in L4 neurons: 31C06-AD, 34G07-DBD>UAS-RSR:PEST, 79C23-S-GS-RSRT-Stop-RSRT-smFP:V5-2a-GAL4, UAS-Ap-shRNA. The genetic element 79C23-S is a bacterial artificial chromosome that encodes the *Brp* gene (*Chen et al., 2014*). 31C06-AD, 34G07-DBD

drives the expression of R recombinase (RSR) in L4 neurons and RSR removes the stop codon from 79C23-S-GS-RSRT-Stop-RSRT-smFP:V5-2a-GAL4. Therefore, Brp:smFP:V5-2a-GAL4 is transcribed and translated in L4 neurons into two proteins, Brp:smFP:V5 and GAL4. GAL4 together with 31C06-AD, 34G07-DBD drives the expression of Ap-shRNA to KD Ap. There is no continuous Ap-shRNA expression when using 31C06-AD, 34G07-DBD to drive expression of Ap-shRNA directly because 31C06-AD, 34G07-DBD depends on Ap.

## Generating Bsh-TaDa fly line

Bsh-TaDa fly line was generated using the FlyORF-TaDa system described in *Aughey et al., 2021*. Homozygous hs-FlpD5; FlyORF-TaDa virgin females were crossed to males from Bsh-ORF-3xHA line. Progeny (larval stage) were heat-shocked at 37°C for 60 min, once per day. After eclosion, F1 male flies were crossed to MKRS/TM6B virgin females. F2 males and virgin females with the correct eye phenotype (w-; 3xP3-dsRed2+) were crossed to establish a balanced stock.

## TaDa in L4 neurons at the time of synapse formation

Homozygous tubP-GAL80[ts]; 31C06-Gal4, UAS-myristoylated-tdTomato males were crossed to homozygous virgin females (FlyORF-TaDa line for Dam; Bsh-TaDa line for Bsh:Dam). Crosses were reared at 18°C. To perform TaDa in L4 neurons during synapse formation window, we collected pupae with the age of 46 hr APF and moved them to 29°C to activate 31C06-Gal4 for 24 hr (*Figure 5—figure supplement 1*). Then lamina were collected (age equivalent at 25°C: 76 hr APF) in cold PBS within 1 hr and stored at –20°C immediately until sufficient lamina were collected – for each group, about 70 lamina from 35 pupae. The TaDa experimental pipeline was followed according to *Marshall et al., 2016*, with a few modifications. Briefly, DNA was extracted using a QIAamp DNA Micro Kit (QIAGEN), digested with DpnI (NEB) overnight, and cleaned up using QIAGEN PCR purification columns. DamID adaptors were ligated using T4 DNA ligase (NEB) followed by DpnII (NEB) digestion for 2 hr and PCR amplification using MyTaq HS DNA polymerase (Bioline). The samples were sequenced on the NovaSeq at 118 base pairs and 27–33 million single end reads per sample.

## Bioinformatic analysis

The TaDa sequencing data was analyzed as described previously (*Sen et al., 2019*). Briefly, each file was assessed for quality using FastQC (v0.11.9). The damidseq pipeline was run to generate Dam bedgraph files, Log2 ratio bedgraph files (Bsh:Dam/Dam), Dam bam files, and Bsh:Dam bam files as described previously (*Marshall and Brand, 2015*). The bedgraph files were used for data visualization on IGV (v.2.13.2) (*Robinson et al., 2011*). The Log2 ratio bedgraph files (Bsh:Dam/Dam) were used for calling Bsh peaks in L4 neurons using find_peaks (FDR < 0.01; min_quant = 0.9) and the generated peak files were used for calling genes using peaks2genes (https://github.com/owenjm/find_peaks, copy archived at *Marshall, 2016*). The gene list of Bsh peaks (score>2.5) in L4 neurons were then combined with L4 scRNAseq data (*Jain et al., 2022*) (normalization number>2). The bam files of Dam and Bsh:Dam were used to create sorted bam files and indexed bam files (bam.bai) using SAMtools (v1.15.1) (*Li et al., 2009*). Sorted bam files and indexed bam files were then computed using the multi-BamSummary and plotCorrelation functions of deepTools (v3.5.1) (*Ramírez et al., 2016*) for correlation coefficients between biological replicates.

## Behavioral experiments

The Fly Vision Box apparatus was developed at the Janelia Research Campus as a high-throughput assay integrating several tests of visually guided behavior for flies in tubes. Groups of flies (usually 10–15) are placed in the clear acrylic tubes, inside a temperature-controlled box. The box contains six tubes, each with a strip of green LEDs (patterns of 4 pixels on/4 pixels off moving at the indicated temporal frequency or contrast) lining one wall, for the motion vision experiments, and a single green and single UV LED on each end of each corridor, for the phototaxis and spectral-preference assays. During the spectral preference task, a green LED is illuminated (at level 10) at the end of each tube, while at the opposite end of the tubes a UV LED is illuminated at increasing brightness levels. The tubes are capped with a polished acrylic plug that is transparent. Four small (pager) motors are mounted on the corners of the box to provide a mechanical startle in between trials. A camera

mounted above the box records fly movements at 25 frames/s. The camera is fitted with an infrared-passing filter and the tubes are suspended above an infrared backlight.

In visual motion response assay, stimulus with different temporal frequency (0, 0.5, 2, 5, 10, 20, 42 Hz) or 5 Hz temporal frequency, but with the indicated contrast level was given. The flies walk against the direction of motion, quantified with a Direction Index. The phototaxis behavior measures the movement of flies toward UV or green LED with the indicated (relative) intensity level at the end of the tubes. The Direction Index is shown integrated over each trial length. In spectral preference task, for lower UV levels, flies walk toward the green LED, but walk toward the UV LED at higher UV levels.

A full experiment lasts ~30 min and consists of 44 conditions, presented in short blocks, during which a series of conditions (e.g. the four LED settings for the phototaxis assay) are presented twice (for phototaxis) or four times each for the other tests, where on consecutive trials the stimulus is presented with opposite direction (for motion) or at the opposite ends of the tubes (for phototaxis and spectral preference). Each trial begins immediately after a 0.5 s mechanical startle by the motors. The box is heated to 34°C. The heating and motorized startle ensure that flies are active throughout and respond to these stimuli.

**Key resources table**

| Reagent type (species) or resource | Designation | Source or reference | Identifiers | Additional information |
|---|---|---|---|---|
| Strain, strain background (*Drosophila melanogaster*) | 10xUAS-IVS-myristoylated-GFP | Bloomington Drosophila Stock Center | RRID: BDSC_32199 | w[1118]; P{y[+t7.7] w[+mC]=10XUAS-IVS-myr::GFP}su(Hw)attP5 |
| Strain, strain background (*D. melanogaster*) | R27G05GAL4 | Bloomington Drosophila Stock Center | RRID: BDSC_48073 | w[1118]; P{y[+t7.7] w[+mC]=GMR27 G05-GAL4}attP2 |
| Strain, strain background (*D. melanogaster*) | UAS-Bsh-RNAi | Bloomington Drosophila Stock Center | RRID: BDSC_29336 | y[1] v[1]; P{y[+t7.7] v[+t1.8]=TRiP.JF02498}attP2 |
| Strain, strain background (*D. melanogaster*) | Bsh-LexA | Bloomington Drosophila Stock Center | RRID: BDSC_52834 | w[1118]; P{y[+t7.7] w[+mC]=GMR64B07-lexA}attP40 |
| Strain, strain background (*D. melanogaster*) | 13xLexAop-IVS-myr::GFP | Bloomington Drosophila Stock Center | RRID: BDSC_32211 | y[1] w[*] P{y[+t7.7] w[+mC]=13XLexAop2-IVS-myr::GFP}su(Hw)attP8 |
| Strain, strain background (*D. melanogaster*) | UAS-Cas9 (attp40) | Bloomington Drosophila Stock Center | RRID: BDSC_58985 | P{ry[+t7.2]=hsFLP}12, y w[*]; P{y[+t7.7] w[+mC]=UAS-Cas9.P2}attP40 |
| Strain, strain background (*D. melanogaster*) | UAS-Cas9 (attp2) | Bloomington Drosophila Stock Center | RRID: BDSC_58986 | P{ry[+t7.2]=hsFLP}12, y[1] w[*]; P{y[+t7.7] w[+mC]=UAS-Cas9.P2}attP2/TM6B, Tb[1] |
| Strain, strain background (*D. melanogaster*) | UAS-Bsh-sgRNA | Vienna Drosophila Resource Center | VDRC 341537 | P{ry[+t7.2]=hsFLP}1, y[1] () w[1118]; P{y[+t7.7] w[+mC]=HD_CFD00611}attP40/CyO-GFP |
| Strain, strain background (*D. melanogaster*) | UAS-Ap-RNAi | Bloomington Drosophila Stock Center | RRID: BDSC_41673 | y[1]sc[*] v[1] sev[21] ; P{y[+t7.7] v[+t1.8]=TRiP.HMS02207}attP2 |
| Strain, strain background (*D. melanogaster*) | 20xUAS-RSR.PEST | Bloomington Drosophila Stock Center | RRID: BDSC_55795 | w[1118]; P{y[+t7.7] w[+mC]=20XUAS-RSR.PEST}attP2 |
| Strain, strain background (*D. melanogaster*) | UAS-Ap-shRNA | Vienna Drosophila Resource Center | VDRC 330463 | P{VSH330463}attP40 |
| Strain, strain background (*D. melanogaster*) | UAS-Bsh-HA | Bloomington Drosophila Stock Center | RRID: BDSC_83310 | y[1] w[1118]; PBac{y[+mDint2] w[+mC]=UAS-bsh.ORF.3xHA.GW}VK00018/CyO |

*Continued on next page*

*Continued*

| Reagent type (species) or resource | Designation | Source or reference | Identifiers | Additional information |
|---|---|---|---|---|
| Strain, strain background (*D. melanogaster*) | tubP-GAL80[ts] | Bloomington Drosophila Stock Center | RRID: BDSC_7017 | w[*]; P{w[+mC]=tubP-GAL80[ts]}2/TM2 |
| Strain, strain background (*D. melanogaster*) | UAS-Zfh1-RNAi | Vienna *Drosophila* Resource Center | VDRC 103205 | P{KK109931}VIE-260B |
| Strain, strain background (*D. melanogaster*) | Brp-rst-stop-rst-smFPV5-2a-GAL4 | Jing Peng (Harvard Medical School) | | W; Bl/CyO-GFP; Brp-rst-stop-rst-smFPV5-2A-Gal4/tm6b |
| Strain, strain background (*D. melanogaster*) | 31C06AD (III), 34G07DBD (III) | Gift from Janelia Research Campus (*Tuthill et al., 2013*) | | w; UAS-FLP/CyO; 31c06A, 34G07DBD/tm6b |
| Strain, strain background (*D. melanogaster*) | 31C06-Gal4, UAS-myristoylated-tdTomato | Gift from Lawrence Zipursky | | ;Bl/CyO; 31c06-Gal4, UAS- myristoylated-tdTomato/tm6b |
| Strain, strain background (*D. melanogaster*) | VALIUM20-mCherry | Bloomington Drosophila Stock Center | RRID: BDSC_35785 | y[1] sc[*]v[1] sev[21]; P{y[+t7.7] v[+t1.8]=VALIUM20-mCherry}attP2 |
| Strain, strain background (*D. melanogaster*) | Bsh-ORF-3XHA (86Fb) | FlyORF Webshop | Cat#F000054 | M{UAS-bsh.ORF.3xHA.GW}ZH-86Fb |
| Strain, strain background (*D. melanogaster*) | flyORF-TaDa | Bloomington Drosophila Stock Center | RRID: BDSC_91637 | w[1118]; M{RFP[3xP3.PB] w[+mC]=FlyORF-TaDa}ZH-86Fb |
| Strain, strain background (*D. melanogaster*) | hs-FlpD5; FlyORF-TaDa | Bloomington Drosophila Stock Center | RRID: BDSC_91638 | w[1118]; P{y[+t7.7] w[+mC]=hs-FLPD5}attP40; M{RFP[3xP3.PB] w[+mC]=FlyORF-TaDa}ZH-86Fb |
| Strain, strain background (*D. melanogaster*) | Bsh-null mutant | Gift from Makoto Sato | | |
| Strain, strain background (*D. melanogaster*) | Bsh-TaDa | This paper | | w; +/CyO; UAS-GFP-Bsh-DAM/tm6b; See Generating Bsh-TaDa fly line in Materials and methods |
| Antibody | Chicken polyclonal | Abcam | Cat#ab13970, RRID_300798 | Anti-GFP (1:1000) |
| Antibody | Rabbit polyclonal | Gift from Claude Desplan (*Özel et al., 2021*) | | Anti-Bsh (1:1000) |
| Antibody | Guinea pig polyclonal | Gift from Lawrence Zipursky (*Tan et al., 2015*) and Makoto Soto | | Anti-Bsh (1:1000) |
| Antibody | Rabbit polyclonal | Gift from Markus Affolter (*Bieli et al., 2015*) | | Anti-Apterous (1:1000) |
| Antibody | Rat monoclonal | Gift from Cheng-Ting Chien (*Chen et al., 2012*) | | Anti-Pdm3 (1:200) |
| Antibody | Rabbit polyclonal | Gift from Cheng-Yu Lee (*Janssens et al., 2014*) | | Anti-Erm (1:100) |

*Continued on next page*

*Continued*

| Reagent type (species) or resource | Designation | Source or reference | Identifiers | Additional information |
|---|---|---|---|---|
| Antibody | Rat monoclonal | Gift from Jing Peng (*Santiago et al., 2021*) | | Anti-Erm (1:70) |
| Antibody | Mouse monoclonal | Developmental Studies Hybridoma Bank | Cat#Seven-up D2D3, RRID_2618079 | Anti-Svp (1:10) |
| Antibody | Rabbit polyclonal | Gift from James Skeath (*Tian et al., 2004*) | | Anti-Zfh1 (1:1000) |
| Antibody | Rabbit polyclonal | Asian Distribution Center for Segmentation Antibodies | Code#812 | Anti-Tailless (1:200) |
| Antibody | Mouse monoclonal | Developmental Studies Hybridoma Bank | Cat#mAbdac1-1, RRID: AB_579773 | Anti-Dac (1:100) |
| Antibody | Mouse monoclonal | Developmental Studies Hybridoma Bank | Cat#Elav-9F8A9, RRID: AB_528217 | Anti-Elav (1:200) |
| Antibody | Mouse monoclonal | Developmental Studies Hybridoma Bank | Cat#Rat-Elav-7E8A10 anti-elav, RRID: AB_528218 | Anti-Elav (1:50) |
| Antibody | Mouse monoclonal | Developmental Studies Hybridoma Bank | Cat#24B10, RRID: AB_528161 | Anti-Chaoptin (1:20) |
| Antibody | Guinea pig polyclonal | Gift from Matthew Pecot Lab (*Xu et al., 2019*) | | Anti-DIP- β (1:300) |
| Antibody | Mouse monoclonal | Bio-Rad Laboratories | Cat#MCA1360A647, RRID: AB_770156 | Anti-V5-TAG:Alexa Fluor 647 (1:300) |
| Antibody | Rat monoclonal | Sigma-Aldrich | Cat#12158167001, RRID: AB_390915 | Anti-HA (1:100) |
| Antibody | Guinea pig polyclonal | Gift from Richard Mann (*Casares and Mann, 1998*) | | Anti-Hth (1:2000) |
| Antibody | Mouse monoclonal | Developmental Studies Hybridoma Bank | Cat#nc-82, RRID: AB_2314866 | Anti-Brp (1:50) |
| Antibody | Donkey polyclonal | Jackson ImmunoResearch Lab | Cat#712-545-153, RRID: AB_2340684 | Alexa Fluor 488 anti-rat (1:400) |
| Antibody | Donkey polyclonal | Jackson ImmunoResearch Lab | Cat#703-545-155, RRID: AB_2340375 | Alexa Fluor 488 anti-chicken (1:400) |
| Antibody | Donkey polyclonal | Jackson ImmunoResearch Lab | Cat#706-545-148, RRID: AB_2340472 | Alexa Fluor 488 anti-guinea pig (1:400) |
| Antibody | Donkey polyclonal | Jackson ImmunoResearch Lab | Cat#711-545-152, RRID: AB_2313584 | Alexa Fluor 488 anti-rabbit (1:400) |

*Continued on next page*

*Continued*

| Reagent type (species) or resource | Designation | Source or reference | Identifiers | Additional information |
|---|---|---|---|---|
| Antibody | Donkey polyclonal | Jackson ImmunoResearch Lab | Cat#715-545-150, RRID: AB_2340846 | Alexa Fluor 488 anti-mouse (1:400) |
| Antibody | Donkey polyclonal | Jackson ImmunoResearch Lab | Cat#715-295-151, RRID: AB_2340832 | Rhodamine Red-X anti-mouse (1:400) |
| Antibody | Donkey polyclonal | Jackson ImmunoResearch Lab | Cat#712-295-153, RRID: AB_2340676 | Rhodamine Red-X anti-rat (1:400) |
| Antibody | Donkey polyclonal | Jackson ImmunoResearch Lab | Cat#711-295-152, RRID: AB_2340613 | Rhodamine Red-X anti-rabbit (1:400) |
| Antibody | Donkey polyclonal | Jackson ImmunoResearch Lab | Cat#706-295-148, RRID: AB_2340468 | Rhodamine Red-X donkey anti-guinea pig (1:400) |
| Antibody | Donkey polyclonal | Jackson ImmunoResearch Lab | Cat#711-605-152, RRID: AB_2492288 | Alexa Fluor 647 donkey anti-rabbit (1:400) |
| Antibody | Donkey polyclonal | Jackson ImmunoResearch Lab | Cat#715-605-151, RRID: AB_2340863 | Alexa Fluor 647 donkey anti-mouse (1:400) |
| Antibody | Donkey polyclonal | Jackson ImmunoResearch Lab | Cat#706-605-148, RRID: AB_2340476 | Alexa Fluor 647 anti-guinea pig (1:400) |
| Antibody | Donkey polyclonal | Jackson ImmunoResearch Lab | Cat#712-605-153, RRID: AB_2340694 | Alexa Fluor 647 anti-rat (1:400) |
| Sequence-based reagent | Oligonucloetide | Integrated DNA Technologies | | DamID Adaptor (top strand): CTAATACGACTCACTATAGGGCAGCGTGGTCGCGGCCGAGGA |
| Sequence-based reagent | Oligonucloetide | Integrated DNA Technologies | | DamID Adaptor (bottom strand): TCCTCGGCCG |
| Sequence-based reagent | Oligonucloetide | Integrated DNA Technologies | | DamID Primer for PCR: GGTCGCGGCCGAGGATC |
| Commercial assay or kit | QIAamp DNA Micro Kit | QIAGEN | Cat#56304 | |
| Commercial assay or kit | PCR Purification Kit | QIAGEN | Cat#28104 | |
| Chemical compound, drug | EDTA | Sigma-Aldrich | Cat#E6758 | |
| Chemical compound, drug | DpnI and CutSmart buffer | NEB | Cat#R0176S | |
| Chemical compound, drug | DpnII and DpnII buffer | NEB | Cat#R0543S | |
| Chemical compound, drug | MyTaq HS DNA Polymerase | Bioline | Cat#BIO-21112 | |

*Continued on next page*

*Continued*

| Reagent type (species) or resource | Designation | Source or reference | Identifiers | Additional information |
|---|---|---|---|---|
| Chemical compound, drug | AlwI | NEB | Cat#R0513S | |
| Chemical compound, drug | RNase A (DNase free) | Roche | Cat#11119915001 | |
| Chemical compound, drug | T4 DNA ligase and 10x buffer | NEB | Cat#M0202S | |
| Software, algorithm | Fiji | *Schindelin et al., 2012* | https://imagej.nih.gov/ij/download.html | |
| Software, algorithm | FastQC (v0.11.9) | The Babraham Bioinformatics group | https://www.bioinformatics.babraham.ac.uk/projects/download.html#fastqc | |
| Software, algorithm | MATLAB | Mathworks | https://www.mathworks.com/products/matlab.html | |
| Software, algorithm | damidseq_pipeline | *Marshall and Brand, 2015* | https://owenjm.github.io/damidseq_pipeline/ | |
| Software, algorithm | Bowtie2 (v2.4.5) | *Langmead and Salzberg, 2012* | http://bowtie-bio.sourceforge.net/bowtie2/index.shtml | |
| Software, algorithm | IGV (v.2.13.2) | *Robinson et al., 2011* | https://software.broadinstitute.org/software/igv/download | |
| Software, algorithm | SAMtools (v1.15.1) | *Li et al., 2009* | http://www.htslib.org/download/ | |
| Software, algorithm | deepTools (v3.5.1) | *Ramírez et al., 2016* | https://deeptools.readthedocs.io/en/develop/content/installation.html | |
| Software, algorithm | Find_peaks | *Marshall et al., 2016* | https://github.com/owenjm/find_peaks | |

## Quantification and statistical analysis for behavioral experiments

Videos are tracked offline with custom code written in MATLAB. After tracking, the trajectories were analyzed on a per-frame basis to see whether flies were walking in one direction (either following the moving patterns or walking toward one end-cap LED) or the other. The behavior is summarized as a 'Direction Index' which is simply the difference between flies walking in one direction and the other (by convention the direction against the motion, or toward the illuminated LED, is the positive one) divided by the total number of flies in the tube. For example, with 12 flies in a tube, during one frame, 7 are moving in the positive direction and 4 are moving in the negative direction (and 1 is not moving), then for this frame, DI = (7–4)/12=0.25.

For the summarized plots in *Figure 7*, the Direction Index is averaged over the entire 10 s duration of each motion stimulus trial. For the phototaxis and spectral preference experiments, the DI is integrated (accumulated) across time and the data recorded for each trial is the peak (negative or positive) of this curve. For the 10 or 15 s trials, the maximum possible value is 10 or 15 (10, 15 s×1). Individual data points in *Figure 7* represent the mean or peak DI metric summarizing all the flies in each tube, and the summary data are the mean and standard deviation across tubes.

The data summarized are part of a larger series of experimental conditions that include several other tasks that are largely redundant with those presented. Nevertheless, the flies experienced 44 different conditions, and so all data were used for a false discovery rate controlling procedure

(*Benjamini and Hochberg, 1995*) with q=0.05. The data from conditions not shown do not contain additional tests with statistically significant differences between the two tested genotypes.

## Quantification of DIP-β fluorescence signal

Using ImageJ, we quantified DIP-β signal in control and KD brains by measuring fluorescence signal along the long axis of lamina cartridges (see white lines in *Figure 6A*) from the distal dash line to the proximal dash line (three cartridges per brain). Signal intensity values and cartridge lengths were converted to percentages by setting the highest intensity within each cartridge as 100% intensity and the full length of the cartridge as 100% distance. Statistical analysis using unpaired t-tests was performed after setting uniform intervals (using the spline function on MATLAB) of 0.01% distance. We presented 20–100% as 0–100% to focus on L4 signal (the first 20% is DIP-β signal in LaWF2).

## Quantification of Brp puncta in the distal and proximal regions of lamina cartridges

Using confocal microscopy, we generated z-stacks of the lamina down the long axis of lamina cartridges. Within each z-stack (i.e. each optic lobe) 20 cartridges in the center of the lamina were identified and the number of Brp puncta in their distal halves was counted. The top (distal edge; top dash line in *Figure 6E*) and bottom (proximal edge; bottom dash line in *Figure 6E*) of each cartridge was determined by the first section below L4 cell body and last section containing L4 neuron processes (myrt-d::TOM), respectively. The midpoint of each cartridge was then identified as the section in between the top and bottom sections. Brp puncta were counted in the sections distal to the midpoint of each cartridge as distal Brp puncta and in the sections proximal to the midpoint of each cartridge as proximal Brp puncta. Genotypes were scored in a blind manner.

## Quantification of L4 proximal neurite length

In the same cartridges as those chosen to quantify the Brp puncta, the number of z-stacks that contained L4 proximal neurites were calculated as a percentage of the entire long axis, and represented the values presented for proximal neurite length.

## Statistical analysis

Statistics were performed using a combination of Microsoft Excel, MATLAB (MathWorks), and Prism (GraphPad) software. Unpaired t-test was used, unless otherwise noted. Data are presented as mean ± SEM unless otherwise noted. A 95% confidence interval was used to define the level of significance. *p<0.05, **p<0.01, ***p<0.001, ns = not significant. All other relevant statistical information can be found in the figure legends.

# Acknowledgements

We thank Stefan Abreo for technical assistance; Claude Desplan, Lawrence Zipursky, Makoto Sato, Markus Affolter, Richard Mann, Jing Peng, Cheng-Yu Lee, James Skeath, Cheng-Ting Chien for antibodies; Emily Nielson for the Fly Vision Box schematic; Claire Managan and Adam Taylor (Janelia Research Campus) for technical assistance and data processing; Lihi Zelnik-Manor and Pietro Perona (Caltech) for the original fly-tracking code; Ayanthi Bhattacharya and Sonia Sen for advice on DamID; and Claude Desplan, Vilaiwan Fernandes, Kristen Lee, Emily Heckman, Peter Newstein, and Sarah Ackerman for comments on the manuscript. Stocks obtained from the Bloomington Drosophila Stock Center were used in this study.

# Additional information

## Funding

| Funder | Grant reference number | Author |
|---|---|---|
| Howard Hughes Medical Institute | | Chris Q Doe |

The funders had no role in study design, data collection and interpretation, or the decision to submit the work for publication.

## Author contributions

Chundi Xu, Conceptualization, Resources, Data curation, Formal analysis, Supervision, Validation, Investigation, Visualization, Methodology, Writing – original draft, Writing – review and editing; Tyler B Ramos, Resources, Data curation, Formal analysis, Validation, Investigation, Writing – review and editing; Edward M Rogers, Resources, Writing – review and editing; Michael B Reiser, Data curation, Software, Formal analysis, Methodology, Writing – review and editing; Chris Q Doe, Resources, Supervision, Funding acquisition, Visualization, Methodology, Project administration, Writing – review and editing

## Author ORCIDs

Chundi Xu ⬤ https://orcid.org/0000-0002-1056-8893
Michael B Reiser ⬤ http://orcid.org/0000-0002-4108-4517
Chris Q Doe ⬤ http://orcid.org/0000-0001-5980-8029

Reviewer #1 (Public Review): https://doi.org/10.7554/eLife.90133.3.sa1
Reviewer #2 (Public Review): https://doi.org/10.7554/eLife.90133.3.sa2
Author Response https://doi.org/10.7554/eLife.90133.3.sa3

# Additional files

## Supplementary files

- MDAR checklist

- Supplementary file 1. Genes that are expressed in L4 neurons and exhibit Bsh binding peaks.

- Supplementary file 2. Genes that are expressed in L4 neurons but do not exhibit Bsh binding peaks.

- Supplementary file 3. Genes that exhibit Bsh binding peaks but are not expressed in L4 neurons.

## Data availability

DamID data in this publication have been deposited in NCBI's GEO and are accessible through GEO Series accession number GSE246726.

The following dataset was generated:

| Author(s) | Year | Dataset title | Dataset URL | Database and Identifier |
|---|---|---|---|---|
| Xu C, Ramos TB, Rogers EM, Reiser MB, Doe CQ | 2023 | Homeodomain proteins hierarchically specify neuronal diversity and synaptic connectivity | https://www.ncbi.nlm.nih.gov/geo/query/acc.cgi?acc=GSE246726 | NCBI Gene Expression Omnibus, GSE246726 |

The following previously published dataset was used:

| Author(s) | Year | Dataset title | Dataset URL | Database and Identifier |
|---|---|---|---|---|
| Jain S, Lin Y, Kurmangaliyev YZ, Zipursky SL | 2021 | A global timing mechanism regulates cell-type specific wiring programs | https://www.ncbi.nlm.nih.gov/geo/query/acc.cgi?acc=GSE190714 | NCBI Gene Expression Omnibus, GSE190714 |

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
