## [Editor Report · eLife assessment]

This paper, offering insights into the mechanisms of neuronal cell type diversification, provides **important** findings that have theoretical or practical implications beyond a single subfield. The data are **compelling** and provide evidence that features methods, data and analyses that are more rigorous than the current state-of-the-art.

---

## [Referee Report · Reviewer #1 (Public Review)]

In this very strong and interesting paper the authors present a convincing series of experiments that reveal molecular mechansism of neuronal cell type diversification in the nervous system of *Drosophila*. The authors show that a homeodomain transcription factor, Bsh, fulfills several critical functions - repressing an alternative fate and inducing downstream homeodomain transcription factors with whom Bsh may collaborate to induce L4 and L5 fates (the author's accompanying paper reveals how Bsh can induce two distinct fates). The authors make elegant use of powerful genetic tools and an arsenal of satisfying cell identity markers.

I believe that this is an important study because it provides some fundamental insights into the conservation of neuronal diversification programs. It is very satisfying to see that similar organizational principles apply in different organism to generate cell type diversity. The authors should also be commended for contextualizing their work very well, giving a broad, scholarly background to the problem of neuronal cell type diversification.

My one suggestion for the authors is to perhaps address in the Discussion (or experimentally address if they wish) how they reconcile that Bsh is on the one hand: (a) continuously expressed in L4/L4, (b) binding directly to a cohort of terminal effectors that are also continuously expressed but then, on the other hand, is not required for their maintaining L4 fate? A few questions: Is Bsh only NOT required for maintaining Ap expression or is it also NOT required for maintaining other terminal markers of L4? The former could be easily explained - Bsh simply kicks of Ap, Ap then autoregulates, but Bsh and Ap then continuously activate terminal effector genes. The second scenario would require a little more complex mechanism: Bsh binding of targets (with Notch) may open chromatin, but then once that's done, Bsh is no longer needed and Ap alone can continue to express genes. I feel that the authors should be at least discussing this. The postmitotic Bsh removal experiment in which they only checked Ap and depression of other markers is a little unsatisfying without further discussion (or experiments, such as testing terminal L4 markers). I hasten to add that this comment does not take away from my overall appreciation for the depth and quality of the data and the importance of their conclusions.

---

## [Referee Report · Reviewer #2 (Public Review)]

Summary:

In this paper, the authors explore the role of the Homeodomain Transcription Factor Bsh in the specification of Lamina neuronal types in the optic lobe of *Drosophila*. Using the framework of terminal selector genes and compelling data, they investigate whether the same factor that establishes early cell identity is responsible for the acquisition of terminal features of the neuron (i.e., cell connectivity and synaptogenesis).

The authors convincingly describe the sequential expression and activity of Bsh, termed here as 'primary HDTF', and of Ap in L4 or Pdm3 in L5 as 'secondary HDTFs' during the specification of these two neurons. The study demonstrates the requirement of Bsh to activate either Ap and Pdm3, and therefore to generate the L4 and L5 fates. Moreover, the authors show that in the absence of Bsh, L4 and L5 fates are transformed into a L1 or L3-like fates.

Finally, the authors used DamID and Bsh:DamID to profile the open chromatin signature and the Bsh binding sites in L4 neurons at the synaptogenesis stage. This allows the identification of putative Bsh target genes in L4, many of which were also found to be upregulated in L4 in a previous single-cell transcriptomic analysis. Among these genes, the paper focuses on Dip-β, a known regulator of L4 connectivity. They demonstrate that both Bsh and Ap are required for Dip-β, forming a feed-forward loop. Indeed, the loss of Bsh causes abnormal L4 synaptogenesis and therefore defects in several visual behaviors.

The authors also propose the intriguing hypothesis that the expression of Bsh expanded the diversity of Lamina neurons from a 3 cell-type state to the current 5 cell-type state in the optic lobe.

Strengths:

Overall, this work presents a beautiful practical example of the framework of terminal selectors: Bsh acts hierarchically with Ap or Pdm3 to establish the L4 or L5 cell fates and, at least in L4, participates in the expression of terminal features of the neuron (i.e., synaptogenesis through Dip-β regulation).

The hierarchical interactions among Bsh and the activation of Ap and Pdm3 expression in L4 and L5, respectively, are well established experimentally. Using different genetic drivers, the authors show a window of competence during L4 neuron specification during which Bsh activates Ap expression. Later, as the neuron matures, Ap becomes independent of Bsh. This allows the authors to propose a coherent and well-supported model in which Bsh acts as a 'primary' selector that activates the expression of L4-specific (Ap) and L5-specific (Pdm3) 'secondary' selector genes, that together establish neuronal fate.

Importantly, the authors describe a striking cell fate change when Bsh is knocked down from L4/L5 progenitor cells. In such case, L1 and L3 neurons are generated at the expense of L4 and L5. The paper demonstrates that Bsh in L4/L5 represses Zfh1, which in turn acts as the primary selector for L1/L3 fates. These results point to a model where the acquisition of Bsh during evolution might have provided the grounds for the generation of new cell types, L4 and L5, expanding lamina neuronal diversity for a more refined visual behaviors in flies. This is an intriguing and novel hypothesis that should be tested from an evo-devo standpoint, for instance by identifying a species when L4 and L5 do not exist and/or Bsh is not expressed in L neurons.

To gain insight into how Bsh regulates neuronal fate and terminal features, the authors have profiled the open chromatin landscape and Bsh binding sites in L4 neurons at mid-pupation using the DamID technique. The paper describes a number of genes that have Bsh binding peaks in their regulatory regions and that are differentially expressed in L4 neurons, based on available scRNAseq data. Although the manuscript does not explore this candidate list in depth, many of these genes belong to classes that might explain terminal features of L4 neurons, such as neurotransmitter identity, neuropeptides or cytoskeletal regulators. Interestingly, one of these upregulated genes with a Bsh peak is Dip-β, an immunoglobulin superfamily protein that has been described by previous work from the author's lab to be relevant to establish L4 proper connectivity. This work proves that Bsh and Ap work in a feed-forward loop to regulate Dip-β expression, and therefore to establish normal L4 synapses. Furthermore, Bsh loss of function in L4 causes impairs visual behaviors.

Weaknesses:

● The last paragraph of the introduction is written using rhetorical questions and does not read well. I suggest rewriting it in a more conventional direct style to improve readability.

● A significant concern is the way in which information is conveyed in the Figures. Throughout the paper, understanding of the experimental results is hindered by the lack of information in the Figure headers. Specifically, the genetic driver used for each panel should be adequately noted, together with the age of the brain and the experimental condition. For example, R27G05-Gal4 drives early expression in LPCs and L4/L5, while the 31C06-AD, 34G07-DBD Split-Gal4 combination drives expression in older L4 neurons, and the use of one or the other to drive Bsh-KD has dramatic differences in Ap expression. The indication of the driver used in each panel will facilitate the reader's grasp of the experimental results.

● Bsh role in L4/L5 cell fate:

o It is not clear whether Tll+/Bsh+ LPCs are the precursors of L4/L5. Morphologically, these cells sit very close to L5, but are much more distant from L4.

o Somatic CRISPR knockout of Bsh seems to have a weaker phenotype than the knockdown using RNAi. However, in several experiments down the line, the authors use CRISPR-KO rather than RNAi to knock down Bsh activity: it should be explained why the authors made this decision. Alternatively, a null mutant could be used to consolidate the loss of function phenotype, although this is not strictly necessary given that the RNAi is highly efficient and almost completely abolishes Bsh protein.

o Line 102: Rephrase "R27G05-Gal4 is expressed in all LPCs and turned off in lamina neurons" to "is turned off as lamina neurons mature", as it is kept on for a significant amount of time after the neurons have already been specified.

o Line 121: "(a) that all known lamina neuron markers become independent of Bsh regulation in neurons" is not an accurate statement, as the markers tested were not shown to be dependent on Bsh in the first place.

o Lines 129-134: Make explicit that the LPC-Gal4 was used in this experiment. This is especially important here, as these results are opposite to the Bsh Loss of Function in L4 neurons described in the previous section. This will help clarify the window of competence in which Bsh establishes L4/L5 neuronal identities through ap/pdm3 expression.

● DamID and Bsh binding profile:

○ Figure 5 - figure supplement 1C-E: The genotype of the Control in (C) has to be described within the panel. As it is, it can be confused with a wild type brain, when it is in fact a Bsh-KO mutant.

○ It Is not clear how L4-specific Differentially Expressed Genes were found. Are these genes DEG between Lamina neurons types, or are they upregulated genes with respect to all neuronal clusters? If the latter is the case, it could explain the discrepancy between scRNAseq DEGs and Bsh peaks in L4 neurons.

● Dip-β regulation:

○ Line 234: It is not clear why CRISPR KO is used in this case, when Bsh-RNAi presents a stronger phenotype.

○ Figure 6N-R shows results using LPC-Gal4. It is not clear why this driver was used, as it makes a less accurate comparison with the other panels in the figure, which use L4-Split-Gal4. This discrepancy should be acknowledged and explained, or the experiment repeated with L4-Split-Gal4>Ap-RNAi.

○ Line 271: It is also possible that L4 activity is dispensable for motion detection and only L5 is required.

● Discussion: It is necessary to de-emphasize the relevance of HDTFs, or at least acknowledge that other, non-homeodomain TFs, can act as selector genes to determine neuronal identity. By restricting the discussion to HDTFs, it is not mentioned that other classes of TFs could follow the same Primary-Secondary selector activation logic.

---

## [Author Response]

The following is the authors’ response to the original reviews.

**Public Reviews:**

**Reviewer #1 (Public Review):**
In this very strong and interesting paper the authors present a convincing series of experiments that reveal molecular mechanism of neuronal cell type diversification in the nervous system of *Drosophila*. The authors show that a homeodomain transcription factor, Bsh, fulfills several critical functions - repressing an alternative fate and inducing downstream homeodomain transcription factors with whom Bsh may collaborate to induce L4 and L5 fates (the author's accompanying paper reveals how Bsh can induce two distinct fates). The authors make elegant use of powerful genetic tools and an arsenal of satisfying cell identity markers.

Thanks!

I believe that this is an important study because it provides some fundamental insights into the conservation of neuronal diversification programs. It is very satisfying to see that similar organizational principles apply in different organisms to generate cell type diversity. The authors should also be commended for contextualizing their work very well, giving a broad, scholarly background to the problem of neuronal cell type diversification.

Thanks!

My one suggestion for the authors is to perhaps address in the Discussion (or experimentally address if they wish) how they reconcile that Bsh is on the one hand: (a) continuously expressed in L4/L4, (b) binding directly to a cohort of terminal effectors that are also continuously expressed but then, on the other hand, is not required for their maintaining L4 fate? A few questions: Is Bsh only NOT required for maintaining Ap expression or is it also NOT required for maintaining other terminal markers of L4? The former could be easily explained - Bsh simply kicks of Ap, Ap then autoregulates, but Bsh and Ap then continuously activate terminal effector genes. The second scenario would require a little more complex mechanism: Bsh binding of targets (with Notch) may open chromatin, but then once that's done, Bsh is no longer needed and Ap alone can continue to express genes. I feel that the authors should be at least discussing this. The postmitotic Bsh removal experiment in which they only checked Ap and depression of other markers is a little unsatisfying without further discussion (or experiments, such as testing terminal L4 markers). I hasten to add that this comment does not take away from my overall appreciation for the depth and quality of the data and the importance of their conclusions.

Great suggestions, we will discuss these two hypotheses as requested.

Bsh initiates Ap expression in L4 neurons which then maintain Ap expression independently of Bsh expression, likely through Ap autoregulation. During the synaptogenesis window, Ap expression becomes independent from Bsh expression, but Bsh and Ap are both still required to activate the synapse recognition molecule DIP-beta. Additionally, Bsh also shows putative binding to other L4 identity genes, e.g., those required for neurotransmitter choice, and electrophysiological properties, suggesting Bsh may initiate L4 identity genes as a suite of genes. The mechanism of maintaining identity features (e.g., morphology, synaptic connectivity, and functional properties) in the adult remains poorly understood. It is a great question whether primary HDTF Bsh maintains the expression of L4 identity genes in the adult. To test this, in our next project, we will specifically knock out Bsh in L4 neurons of the adult fly and examine the effect on L4 morphology, connectivity, and function properties.

**Reviewer #2 (Public Review):**
Summary:In this paper, the authors explore the role of the Homeodomain Transcription Factor Bsh in the specification of Lamina neuronal types in the optic lobe of *Drosophila*. Using the framework of terminal selector genes and compelling data, they investigate whether the same factor that establishes early cell identity is responsible for the acquisition of terminal features of the neuron (i.e., cell connectivity and synaptogenesis).

Thanks for the positive words!

The authors convincingly describe the sequential expression and activity of Bsh, termed here as 'primary HDTF', and of Ap in L4 or Pdm3 in L5 as 'secondary HDTFs' during the specification of these two neurons. The study demonstrates the requirement of Bsh to activate either Ap and Pdm3, and therefore to generate the L4 and L5 fates. Moreover, the authors show that in the absence of Bsh, L4 and L5 fates are transformed into a L1 or L3-like fates.

Thanks!

Finally, the authors used DamID and Bsh:DamID to profile the open chromatin signature and the Bsh binding sites in L4 neurons at the synaptogenesis stage. This allows the identification of putative Bsh target genes in L4, many of which were also found to be upregulated in L4 in a previous single-cell transcriptomic analysis. Among these genes, the paper focuses on Dip-β, a known regulator of L4 connectivity. They demonstrate that both Bsh and Ap are required for Dip-β, forming a feed-forward loop. Indeed, the loss of Bsh causes abnormal L4 synaptogenesis and therefore defects in several visual behaviors. The authors also propose the intriguing hypothesis that the expression of Bsh expanded the diversity of Lamina neurons from a 3 cell-type state to the current 5 cell-type state in the optic lobe.

Thanks for the excellent summary of our findings!

Strengths:Overall, this work presents a beautiful practical example of the framework of terminal selectors: Bsh acts hierarchically with Ap or Pdm3 to establish the L4 or L5 cell fates and, at least in L4, participates in the expression of terminal features of the neuron (i.e., synaptogenesis through Dip-β regulation).

Thanks!

The hierarchical interactions among Bsh and the activation of Ap and Pdm3 expression in L4 and L5, respectively, are well established experimentally. Using different genetic drivers, the authors show a window of competence during L4 neuron specification during which Bsh activates Ap expression. Later, as the neuron matures, Ap becomes independent of Bsh. This allows the authors to propose a coherent and well-supported model in which Bsh acts as a 'primary' selector that activates the expression of L4specific (Ap) and L5-specific (Pdm3) 'secondary' selector genes, that together establish neuronal fate.

Thanks again!

Importantly, the authors describe a striking cell fate change when Bsh is knocked down from L4/L5 progenitor cells. In such cases, L1 and L3 neurons are generated at the expense of L4 and L5. The paper demonstrates that Bsh in L4/L5 represses Zfh1, which in turn acts as the primary selector for L1/L3 fates. These results point to a model where the acquisition of Bsh during evolution might have provided the grounds for the generation of new cell types, L4 and L5, expanding lamina neuronal diversity for a more refined visual behaviors in flies. This is an intriguing and novel hypothesis that should be tested from an evo-devo standpoint, for instance by identifying a species when L4 and L5 do not exist and/or Bsh is not expressed in L neurons.

Thanks for the appreciation of our findings!

To gain insight into how Bsh regulates neuronal fate and terminal features, the authors have profiled the open chromatin landscape and Bsh binding sites in L4 neurons at mid-pupation using the DamID technique. The paper describes a number of genes that have Bsh binding peaks in their regulatory regions and that are differentially expressed in L4 neurons, based on available scRNAseq data. Although the manuscript does not explore this candidate list in depth, many of these genes belong to classes that might explain terminal features of L4 neurons, such as neurotransmitter identity, neuropeptides or cytoskeletal regulators. Interestingly, one of these upregulated genes with a Bsh peak is Dip-β, an immunoglobulin superfamily protein that has been described by previous work from the author's lab to be relevant to establish L4 proper connectivity. This work proves that Bsh and Ap work in a feed-forward loop to regulate Dip-β expression, and therefore to establish normal L4 synapses. Furthermore, Bsh loss of function in L4 causes impairs visual behaviors.Thanks for the excellent summary of our findings.Weaknesses:● The last paragraph of the introduction is written using rhetorical questions and does not read well. I suggest rewriting it in a more conventional direct style to improve readability.

We agree and have updated the text as suggested.

● A significant concern is the way in which information is conveyed in the Figures. Throughout the paper, understanding of the experimental results is hindered by the lack of information in the Figure headers. Specifically, the genetic driver used for each panel should be adequately noted, together with the age of the brain and the experimental condition. For example, R27G05-Gal4 drives early expression in LPCs and L4/L5, while the 31C06-AD, 34G07-DBD Split-Gal4 combination drives expression in older L4 neurons, and the use of one or the other to drive Bsh-KD has dramatic differences in Ap expression. The indication of the driver used in each panel will facilitate the reader's grasp of the experimental results.

We agree and have updated the figure annotation.

● Bsh role in L4/L5 cell fate: o It is not clear whether Tll+/Bsh+ LPCs are the precursors of L4/L5. Morphologically, these cells sit very close to L5, but are much more distant from L4.

Our current data show L4 and L5 neurons are generated by different LPCs. However, currently, we don’t have tools to demonstrate which subset of LPCs generate which lamina neuron type. We are currently working on a follow-up manuscript on LPC heterogeneity, but those experiments have just barely been started.

● Somatic CRISPR knockout of Bsh seems to have a weaker phenotype than the knockdown using RNAi. However, in several experiments down the line, the authors use CRISPR-KO rather than RNAi to knock down Bsh activity: it should be explained why the authors made this decision. Alternatively, a null mutant could be used to consolidate the loss of function phenotype, although this is not strictly necessary given that the RNAi is highly efficient and almost completely abolishes Bsh protein.

The reason we chose CRISPR-KO (L4-specific Gal4, uas-Cas9, and uas-Bsh-sgRNAs) is that it effectively removed Bsh expression from the majority of L4 neurons. However, it failed to knock down Bsh in L4 neurons using L4-split Gal4 and Bsh-RNAi because L4-split Gal4 expression depends on Bsh. We have updated this explanation in the text.

● Line 102: Rephrase "R27G05-Gal4 is expressed in all LPCs and turned off in lamina neurons" to "is turned off as lamina neurons mature", as it is kept on for a significant amount of time after the neurons have already been specified.

Thanks; we have made that change.

● Line 121: "(a) that all known lamina neuron markers become independent of Bsh regulation in neurons" is not an accurate statement, as the markers tested were not shown to be dependent on Bsh in the first place.

Good point. We have rephrased it as “that all known lamina neuron markers are independent of Bsh regulation in neurons”.

● Lines 129-134: Make explicit that the LPC-Gal4 was used in this experiment. This is especially important here, as these results are opposite to the Bsh Loss of Function in L4 neurons described in the previous section. This will help clarify the window of competence in which Bsh establishes L4/L5 neuronal identities through ap/pdm3 expression.

Thanks! We have updated Gal4 information in the text for every manipulation.

● DamID and Bsh binding profile:● Figure 5 - figure supplement 1C-E: The genotype of the Control in (C) has to be described within the panel. As it is, it can be confused with a wild type brain, when it is in fact a Bsh-KO mutant.

Great point! Thank you for catching this and we have updated it.

● It Is not clear how L4-specific Differentially Expressed Genes were found. Are these genes DEG between Lamina neurons types, or are they upregulated genes with respect to all neuronal clusters? If the latter is the case, it could explain the discrepancy between scRNAseq DEGs and Bsh peaks in L4 neurons.

We did not use “L4-specific Differentially Expressed Genes”. Instead, we used all genes that are significantly transcribed in L4 neurons (line 209-213).

● Dip-β regulation:● Line 234: It is not clear why CRISPR KO is used in this case, when Bsh-RNAi presents a stronger phenotype.

As we explained above, the reason we chose CRISPR-KO (L4-specific Gal4, uas-Cas9, and uas-BshsgRNAs) is that it effectively removed Bsh expression from the majority of L4 neurons. However, it failed to knock down Bsh in L4 neurons using L4-split Gal4 and Bsh-RNAi because L4-split Gal4 expression depends on Bsh. We have updated this explanation in the text.

● Figure 6N-R shows results using LPC-Gal4. It is not clear why this driver was used, as it makes a less accurate comparison with the other panels in the figure, which use L4-Split-Gal4. This discrepancy should be acknowledged and explained, or the experiment repeated with L4-Split-Gal4>Ap-RNAi.

I think you mean 6J-M shows results using LPC-Gal4. We first tried L4-Split-Gal4>Ap-RNAi but it failed to knock down Ap because L4-Split-Gal4 expression depends on Ap. We have added this to the text.

● Line 271: It is also possible that L4 activity is dispensable for motion detection and only L5 is required.

Thanks! Work from Tuthill et al, 2013 showed that L5 is not required for any motion detection. We have included this citation in the text.

● Discussion: It is necessary to de-emphasize the relevance of HDTFs, or at least acknowledge that other, non-homeodomain TFs, can act as selector genes to determine neuronal identity. By restricting the discussion to HDTFs, it is not mentioned that other classes of TFs could follow the same PrimarySecondary selector activation logic.

That is a great point, thank you! We have included this in the discussion.